# Radiogenic strontium isotope variability in the Valley of Oaxaca: A predictive isoscape for Mesoamerican paleomobility studies

Sofía I. Pacheco-Forés[1]*, Nicolas Gauthier[2]*, Lacey B. Carpenter[3], Gwyneth Gordon[4], Kelly J. Knudson[5]

1 Department of Anthropology, University of Minnesota, Minneapolis, Minnesota, United States of America,
2 Florida Museum of Natural History, University of Florida, Gainesville, Florida, United States of America,
3 Department of Anthropology, State University of New York, Buffalo, New York, United States of America,
4 School of Earth and Space Exploration, Arizona State University, Tempe, Arizona, United States of America, 5 Center for Bioarchaeological Research, School of Human Evolution and Social Change, Arizona State University, Tempe, Arizona, United States of America

☯ These authors contributed equally to this work.
* sipf@umn.edu (SIPF); nicolas.gauthier@ufl.edu (NG)

## Abstract

Radiogenic strontium ($^{87}Sr/^{86}Sr$) isotope analysis is a well-established method for reconstructing the mobility of human populations in the past and present. Baseline $^{87}Sr/^{86}Sr$ data are fundamental to the method, as Sr varies across the landscape according to local geology and geoenvironmental factors. The method's application within studies of ancient Mesoamerican paleomobility, however, has concentrated on two key regions—Teotihuacan and the Maya region—despite its potential broader relevance across greater Mesoamerica. This is due in part to a lack of available baseline $^{87}Sr/^{86}Sr$ data for the region at large. Using the Valley of Oaxaca as a case study, we use Bayesian Additive Regression Trees (BART) to generate a locally calibrated predictive $^{87}Sr/^{86}Sr$ isoscape model of Mesoamerica in general and the Valley of Oaxaca in particular. We integrate (1) observed $^{87}Sr/^{86}Sr$ data from modern plant samples ($n = 95$) from 17 sites across the Valley, (2) a compiled database of continental North and South American $^{87}Sr/^{86}Sr$ data, (3) geological bedrock maps, and (4) high resolution spatial data on geoenvironmental Sr covariates to iteratively develop and test a high performing predictive model for Mesoamerica, highlighting the importance of regional calibration in developing predictive $^{87}Sr/^{86}Sr$ isoscapes. Our results indicate that though overlap exists, $^{87}Sr/^{86}Sr$ can be used to detect migration within the Valley of Oaxaca as well as between the Valley and greater Mesoamerica. We then apply our isoscape to previously published human $^{87}Sr/^{86}Sr$ data from Monte Albán, Oaxaca to demonstrate how our model's explicit quantification of uncertainty in local $^{87}Sr/^{86}Sr$ ranges allows for more nuanced interpretation of paleomobility in archaeological samples.

**Data availability statement:** All relevant data are within the manuscript. Associated code is available via Zenodo: https://github.com/flmnh-ai/oaxaca-isoscape.

**Funding:** This research was funded by the National Science Foundation (2013155229 and 1744335 awarded to SIPF; 2409068 awarded to NG and SIPF). The funders had no role in study design, data collection and analysis, decision to publish, or preparation of the manuscript.

**Competing interests:** The authors have declared that no competing interests exist.

## Introduction

There is a long tradition of archaeological and bioarchaeological research in the Valley of Oaxaca [1–5]. Much of this research focuses on the emergence of one of the earliest state-level societies in the Americas—the multi-ethnic Zapotec state of Monte Albán in the Valley of Oaxaca of southern Mexico during the Late Formative period (300−100 BCE). Archaeologists have documented marked shifts in material culture and settlement patterns during this time period, leading them to infer that both regional and interregional migration may have played a role in the development of the Zapotec state [6]. Although the last 40 years have seen an explosion in biogeochemical studies of human skeletal remains to directly test Mesoamerican migration hypotheses [7], these paleomobility studies have focused overwhelmingly on central Mexico and the central Maya lowlands [8]. Very few biogeochemical studies of migration exist in the Valley of Oaxaca [but see 9–12]. This dearth of biogeochemical investigations of migration in the Valley of Oaxaca may be due at least in part to the lack of published isotopic baseline data as well as the geologic complexity of the region.

The development of spatially predictive maps of environmental isotopic variation called isoscapes has greatly aided paleomobility research in regions lacking isotope baseline data [13]. In this article, we examine radiogenic strontium ($^{87}Sr/^{86}Sr$) variability across the Valley of Oaxaca. We present $^{87}Sr/^{86}Sr$ values from a total of 95 modern plant samples from 17 sites across the Valley of Oaxaca. We use the resulting dataset, together with previously compiled $^{87}Sr/^{86}Sr$ data, to develop and test a predictive isoscape model tuned to Mesoamerica in general and the Valley of Oaxaca in particular. Our goal is to provide a high performing predictive $^{87}Sr/^{86}Sr$ baseline isoscape for use in paleomobility studies in the Valley of Oaxaca.

### Assessing migration through biogeochemistry in the Valley of Oaxaca

The Valley of Oaxaca is made up of a series of three interconnected valleys forming a triangular Y-shaped area (Fig 1). The Etla arm extends to the northwest, the Tlacolula arm extends east, and the Ocotlán-Zimatlán arm extends to the south. During the Rosario (700−500 BCE) phase of the Middle Formative period, three major settlements—San José Mogote, Yeguih, and San Martín Tilcajete—emerged in each of the three branches of the Valley of Oaxaca. Monte Albán was founded in the center of the Valley at the beginning of the Early Monte Albán I phase (500–300 BCE) [1,3,14]. Monte Albán continued to expand its influence through the Late Monte Albán I phase (300 BCE – 100 BCE) and the Monte Albán II phase (100 BCE – 200 CE) [15,16].

Both regional and interregional migration played a crucial role in the models for the sociopolitical development of the Valley of Oaxaca. Marked population growth and demographic shifts within the Valley coinciding with the founding of Monte Albán have been attributed to the arrival of migrants into the region as well as the movement of peoples within the Valley itself [1,3,19–24]. Migration has also been implicated in competing population centers' resistance to Monte Albán's expansionism [16,25]. Following Monte Albán's consolidation of power, migration was essential to the state's administrative strategies and trade networks [9,26–28]. Despite this, most

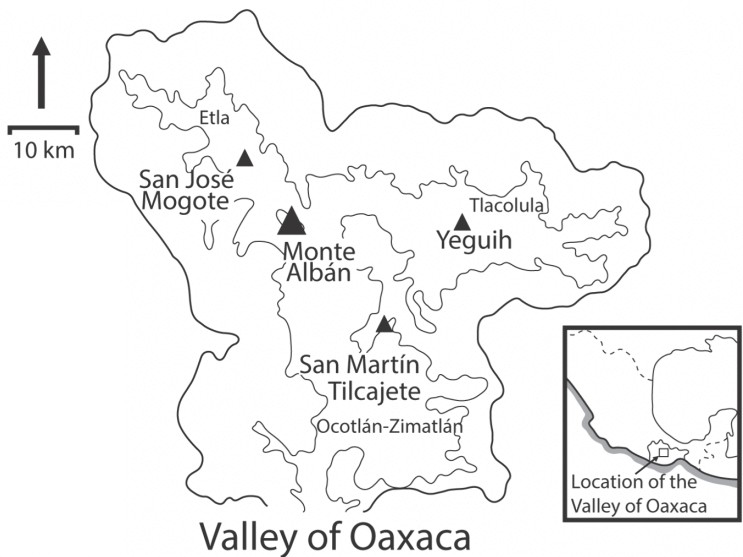
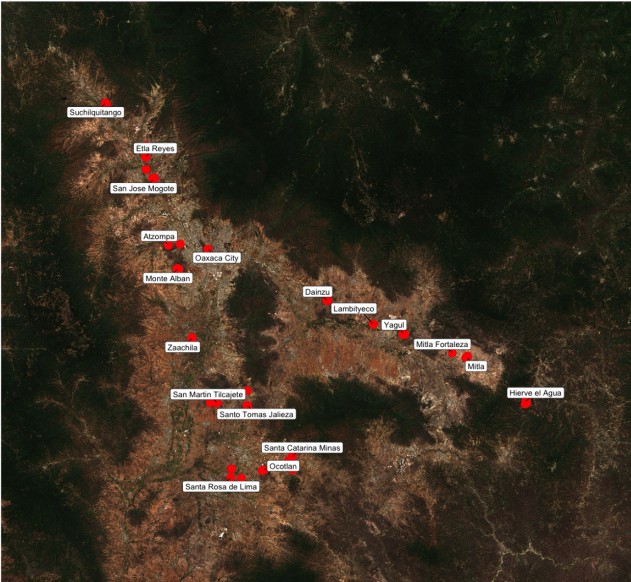

**Fig 1. The Valley of Oaxaca.** The three branches of the Valley of Oaxaca and their corresponding regional centers (left) alongside a map of the 17 sites in the Valley of Oaxaca from which we collected modern plants for $^{87}Sr/^{86}Sr$ analysis (right). Left map modified and reprinted from [17] under a CC BY license, with permission from Carpenter, original copyright 2019. Right map created with publicly available satellite imagery from Copernicus Sentinel-2 data [18] via Google Earth Engine.

archaeological evidence of these migrations remains indirect, implied through settlement patterns, population estimates, and stylistic similarities in architecture and material culture. These data sources can offer insights into the role migration played in the region's cultural development, but the most direct way to evaluate this is to first identify the migrants themselves.

Biogeochemical methods allow archaeologists to directly investigate migration on an individual scale. Hydrogen ($\delta^2H$), stable oxygen ($\delta^{18}O$), sulfur ($\delta^{34}S$), radiogenic strontium ($^{87}Sr/^{86}Sr$), and lead ($^{206/204}Pb$, $^{207/204}Pb$, $^{208/204}Pb$, $^{207/206}Pb$) isotope analyses provide archaeologists with well-established means of reconstructing the residential histories of past peoples [e.g., 29–33]. These isotope systems vary geographically. Hydrogen and oxygen isotopes vary according to regional hydrology and reflect environmental factors such as elevation, temperature, humidity, and latitude [29,34–37]. In contrast, sulfur, radiogenic strontium, and lead isotope ratios vary according to local geology, reflecting the age and composition of geologic bedrock [38–42]. These isotopes are incorporated into human calcified tissues via consumed food, imbibed liquids, and inhaled dust particles and will reflect the isotopic signature of the region where an individual lived during tissue development [36,43–49]. Archaeologists can thus detect individual migrants by identifying changes in isotopic signatures between tissues forming at different times over the life course and/or divergences between isotopic signatures in these tissues and the burial environment, as these differences would signify residence in isotopically distinct regions [e.g., 10,50].

While biogeochemical analyses of paleomobility are well established and have been very successful in detecting migration across ancient Mesoamerica, most studies focus on the central Mexican city of Teotihuacan and the Maya region [8]. Only a handful of studies to date have attempted to directly identify migrants in the Valley of Oaxaca, and all but one of these have focused exclusively on Monte Albán (Table 1). Cumulatively, the four studies examine paleomobility among 75 individuals in the Valley of Oaxaca [9–12]. Only 8% of these individuals were identified as migrants. This limited evidence of migration is surprising, as the preceding discussion indicates there is ample archaeological evidence of both regional and interregional migration in the Valley of Oaxaca.

Table 1. Biogeochemical studies of migration in the Valley of Oaxaca[a].

| Site | Time period | Method | n migrants identified/ N sampled individuals | Data source |
|---|---|---|---|---|
| Monte Albán | 500 BCE – 1520 CE | $\delta^{18}O$ in bone phosphate | 0/16 | [12] |
| Monte Albán | 100 BCE – 550 CE | $^{87}Sr/^{86}Sr$ in enamel and bone apatite | 1/5 | [10] |
| Monte Albán | 100 BCE – 1350 CE | $\delta^{18}O$ in enamel carbonate | 5/38 | [9] |
| Teposcolula Yucundaa | 1295-1634 CE | $\delta^{18}O$ in enamel carbonate | 0/16 | [11] |

[a]This table only includes studies of individuals excavated from secure archaeological contexts. It does not include data from Olivares Flores [51], as that analysis uses $\delta^{18}O$ analysis of enamel carbonate from $n = 7$ incised skulls of unknown provenience to argue that six of the seven individuals likely originate from the Valley of Oaxaca.

## Identifying migrants using biogeochemical baselines and isoscapes

The scarcity of biogeochemically-identified migrants in the Valley of Oaxaca despite the long history of archaeological investigation in the region and its status as one of the first multi-ethnic state-level societies to emerge in the Americas may be due to the lack of published biogeochemical baseline data. Biogeochemical baselines are essential in establishing expectations for characterizing "local" isotopic signatures [52]. If an individual's observed isotopic values diverge sufficiently from a "local" biogeochemical baseline, the individual is designated a "non-local" and thus most likely a migrant. These biogeochemical baselines are created through the analysis of environmental (typically plant or faunal) samples from a site of interest and the application of basic summary statistics to establish thresholds for local $^{87}Sr/^{86}Sr$ variation [38,52]. Archaeologists have used the mean $^{87}Sr/^{86}Sr$ value of environmental samples from a site ± two standard deviations [53,54] or their interquartile range [55] to create biogeochemical baselines defining local $^{87}Sr/^{86}Sr$ variation at a particular site.

None of the previous biogeochemical studies of paleomobility in the Valley of Oaxaca, however, employ biogeochemical baselines. Instead, they identify statistical outliers among the observed human values as possible migrants. While there is an abundance of published environmental isotopic data for other regions of Mexico [e.g., 56,57], such data are scarce for the Valley of Oaxaca. This makes the creation of relevant regional biogeochemical baselines difficult. Furthermore, although biogeochemical baselines provide an objective means of inferring an individual's mobility status relative to expected local $^{87}Sr/^{86}Sr$ values, the statistical means of establishing thresholds to delimit local variation is arbitrary [52] and does not account for the non-normal distribution of $^{87}Sr/^{86}Sr$ values across the landscape [55]. As such, the studies [9–12] cited in Table 1 represent a promising start to investigating the role of migration in the cultural development of the Valley of Oaxaca and the Zapotec state, but more work remains to be done.

**Isoscapes and paleomobility research in the Valley of Oaxaca.** In response to the uneven regional coverage of published biogeochemical baseline data and the limitations of threshold approaches to biogeochemical baselines, researchers have increasingly adopted the use of isoscapes to fill in the blanks. Isoscapes are spatially predictive maps of environmental isotopic variation [13]. Some isoscapes are generated through the geostatistical interpolation (e.g., kriging) of observed isotopic data [58–62]. Geostatistical isoscapes are based on spatial autocorrelation, the idea that points geographically closer to each other on a map tend to be environmentally (for H and O) or geologically (for S, Sr, and Pb) similar. In addition to using empirical isotopic data, these geostatistical models can incorporate covariates such as hydrological or geological maps to further constrain predicted isotope variability. Because isotopes are often non-normally distributed across the landscape, however, these geostatistical isoscapes may falsely represent the actual variability in the selected isotope system [63]. As such, they must be used with caution.

Geostatistical isoscapes using empirical $\delta^{18}O$ values from water samples taken across Mexico show the predicted range of clinal $\delta^{18}O$ variability across Mexico [64,65]. Moreiras Reynaga and colleagues [66] divide this clinal variation into five distinct or slightly overlapping zones of $\delta^{18}O$ values. The zone containing the Valley of Oaxaca covers a large swathe

of western Mexico, including most of Mexico's Pacific Coast [66]. The fact that such a massive geographical area shares similar $\delta^{18}O$ values means there is a high probability of missing potential migrants in the Valley of Oaxaca from other parts of the Pacific Coast. The inability of $\delta^{18}O$ analyses to address questions of regional mobility within the Valley of Oaxaca and their limited utility in assessing interregional mobility across much of western Mexico has led some scholars to posit that $\delta^{18}O$ isotopes may not have a sufficient range of variation to detect the presence of non-local individuals within the Valley [9,51]. This may explain why three of the four existing paleomobility studies in the Valley of Oaxaca using $\delta^{18}O$ identify so few migrants (Table 1). Targeting other isotope systems with greater variability in the Valley of Oaxaca and/or using multiple isotope systems rather than relying on a single system present the best path forward for biogeochemical studies of migration in the Valley of Oaxaca.

Mechanistic models represent another approach to generating isoscapes. This method employs first principles of isotope geochemistry to predict the evolution of a targeted isotope system in a given region [67]. The model's predictions are then compared to published isotope baseline data to test its performance [68–71]. While promising, mechanistic model isoscapes are limited by insufficiently detailed, inaccurate, or inconsistent geologic maps that in turn produce poorly performing models. In addition, estimates from geological maps must take into consideration soil weathering, plant rooting depth and vertical soil variation, and the relative bioavailability of elements which will cause significant modification of strontium isotopes between underlying lithology and the peoples that live in the region. Soil weathering and vertical soil variation are strongly dependent on climate. The interplay between these controlling parameters is poorly understood, making predictive models from purely mechanistic factors problematic.

The challenges posed by both empirical geostatistical and mechanistic model isoscapes have led to the emergence of a hybrid process-based statistical approach to creating isoscapes. This method uses random forest regression, a decision tree-based machine learning algorithm. A decision tree works by repeatedly splitting data based on predictor values (such as bedrock age or rainfall) to create increasingly homogeneous groups. For example, a simple tree might first divide samples based on bedrock age (older or younger than 100 million years), then further divide each group based on elevation, and so on until it can accurately predict strontium values. Decision trees are intuitive but can be unstable and prone to overfitting when used alone. Random forests and similar methods address this limitation by combining hundreds of trees, each considering different combinations of predictors. This ensemble approach produces more robust predictions than a single tree. Hybrid process-based isoscapes incorporate empirical isotope baseline data along with other geoenvironmental covariates used in geostatistical isoscapes into the mechanistic modeling framework to predict isotope values [63,67,72,73]. This hybrid approach simultaneously overcomes limitations in both geostatistical and mechanistic isoscape models, yielding higher performing models that accurately account for non-normal distributions of isotopes across targeted regions of interest.

Quantifying the uncertainty of isoscape predictions is essential for distinguishing between local and non-local individuals in archaeological contexts [74]. Isoscapes generated using random forest regression, however, do not inherently provide uncertainty estimates for their predictions of local isotope values. Quantile regression forests can be used to address this shortcoming, allowing researchers to generate spatially explicit maps of uncertainty to accompany predictive isoscapes [63,75,76]. This method estimates conditional quantiles based on the distribution of outcomes from similar observations across all trees in the forest, providing estimates of the uncertainty associated with the prediction of a particular isotopic value [77]. However, this approach may underestimate uncertainty in regions with sparse training data, such as geologically complex and under-sampled regions like the Valley of Oaxaca.

**Strontium isotopes and the Valley of Oaxaca.** The Valley of Oaxaca represents a particularly attractive candidate for a process-based statistical $^{87}Sr/^{86}Sr$ isoscape. Bioavailable radiogenic strontium varies according to the age and composition of local bedrock, but factors such as the erosion and preferential weathering of bedrock, as well as the addition of material from windblown dust and sea spray can significantly alter $^{87}Sr/^{86}Sr$ values in the environment that are ultimately incorporated into human tissues [for an in-depth discussion of Sr systematics and bioavailable $^{87}Sr/^{86}Sr$ cycling within ecosystems, see [38,39,78].

The Valley of Oaxaca is located in the Sierra Madre Sur, the most geologically complex and poorly understood mor-photectonic province of Mexico [79]. This province contains some of the oldest metamorphic geological formations in Mexico—Precambrian granulites dubbed the Oaxacan Complex that developed from a continental rift sequence [80]. This deeply eroded Precambrian terrain is unconformably overlain by isolated Paleozoic and Mesozoic granitoids and forms the western boundary of the Valley of Oaxaca. While the Valley floor is comprised of recent Quaternary alluvial deposits, the Valley's northeastern edge is made up of Cambrian-to-Eocene mudstone-sandstone rocks bisected by the Oaxaca Fault and a series of Paleozoic-Jurassic metamorphic rocks called the Sierra de Juarez Complex. A mix of Cenozoic volcanic andesitic and magmatic rhyolitic tuffs, along with Mesozoic limestone-dolostone carbonates make up the Valley's southeastern border. The Valley's southern border is dominated by Cenozoic andesitic-rhyolitic tuff, interspersed with Mesozoic carbonates and Cenozoic granitoids intrusive outcrops [79,81–85]. The marked variability in the age and composition of the underlying bedrock which provides the source material for the Quaternary alluvial deposits in and surrounding the Valley of Oaxaca thus promises a high level of intra- and interregional variation in $^{87}Sr/^{86}Sr$ isotopes (Fig 2).

Although no hybrid isoscapes for any isotope system exist in the Valley of Oaxaca, Bataille and colleagues [67] present a $^{87}Sr/^{86}Sr$ isoscape model that uses random forest regression to predict bioavailable $^{87}Sr/^{86}Sr$ variability globally. Their groundbreaking model integrates compiled empirical $^{87}Sr/^{86}Sr$ data and auxiliary covariates known to affect bioavailable $^{87}Sr/^{86}Sr$ with a mechanistic model framework based on geological bedrock. Bataille and colleagues test the model using empirical $^{87}Sr/^{86}Sr$ data. The model performs well in predicting observed $^{87}Sr/^{86}Sr$ values in data-rich regions but performs poorly when predicting observed $^{87}Sr/^{86}Sr$ values in under-sampled regions with complex geologies. The authors conclude that the addition of just tens to hundreds of $^{87}Sr/^{86}Sr$ data points in under-sampled geologically complex areas significantly improves the regional accuracy of the global model. Moreover, in some cases, a regionally specific isoscape model eschewing global bioavailable $^{87}Sr/^{86}Sr$ data is necessary to overcome such models' strong predictive bias towards data-rich regions.

A regionally specific isoscape model could be invaluable in furthering paleomobility research in the Valley of Oaxaca. Unfortunately, very little empirical $^{87}Sr/^{86}Sr$ baseline data in the Valley of Oaxaca exist. Price and colleagues provide some site-level averages of $^{87}Sr/^{86}Sr$ values, but the entire Valley is represented by just 10 samples from five sites, and no raw data are published [55]. To address this gap in the research, we present $^{87}Sr/^{86}Sr$ baseline data for 95 modern plant samples from 17 sites across the Valley of Oaxaca. We use this dataset, along with a compiled database of continental North and South American $^{87}Sr/^{86}Sr$ data, to iteratively train and test a regional Mesoamerican $^{87}Sr/^{86}Sr$ isoscape model focused on the Valley of Oaxaca. Our goal is to provide a high-quality locally calibrated predictive $^{87}Sr/^{86}Sr$ isoscape that will lay the groundwork for increased biogeochemical investigation of paleomobility in the Valley of Oaxaca.

## Materials and methods

### Sample collection

We opportunistically collected a total of 95 modern plant samples from 17 archaeological sites across the Valley of Oaxaca, with an average of five plant samples per site. We targeted plants for baseline sampling as $^{87}Sr/^{86}Sr$ in plants reflect a more consistent average of the local bioavailable $^{87}Sr/^{86}Sr$ in a given ecosystem than whole soils, which can vary greatly even in a small area due to the distinct strontium concentrations and weathering profiles of minerals in the underlying bedrock [38]. Even so, plants with varied rooting depths have access to different sources of Sr (i.e., bedrock vs heavily weathered soil horizons vs wind-borne dust), introducing variation to their observed $^{87}Sr/^{86}Sr$ values [88]. As such, we sampled both shallow and deep rooting depths whenever possible to account for these different local $^{87}Sr/^{86}Sr$ sources. Furthermore, to better simulate sources of $^{87}Sr/^{86}Sr$ in prehispanic diets, we prioritized sampling plant taxa that contributed to ancient diets in the Valley of Oaxaca, such as maguey (*Agave* spp.) and prickly pears (*Opuntia ficus*), or plant taxa with similar rooting depths to ancient staples such as maize (*Zea mays*) that we were unable to sample [89,90]. Additionally, we sought to avoid sampling plants treated with fertilizers or irrigation water, as these could skew signatures of local

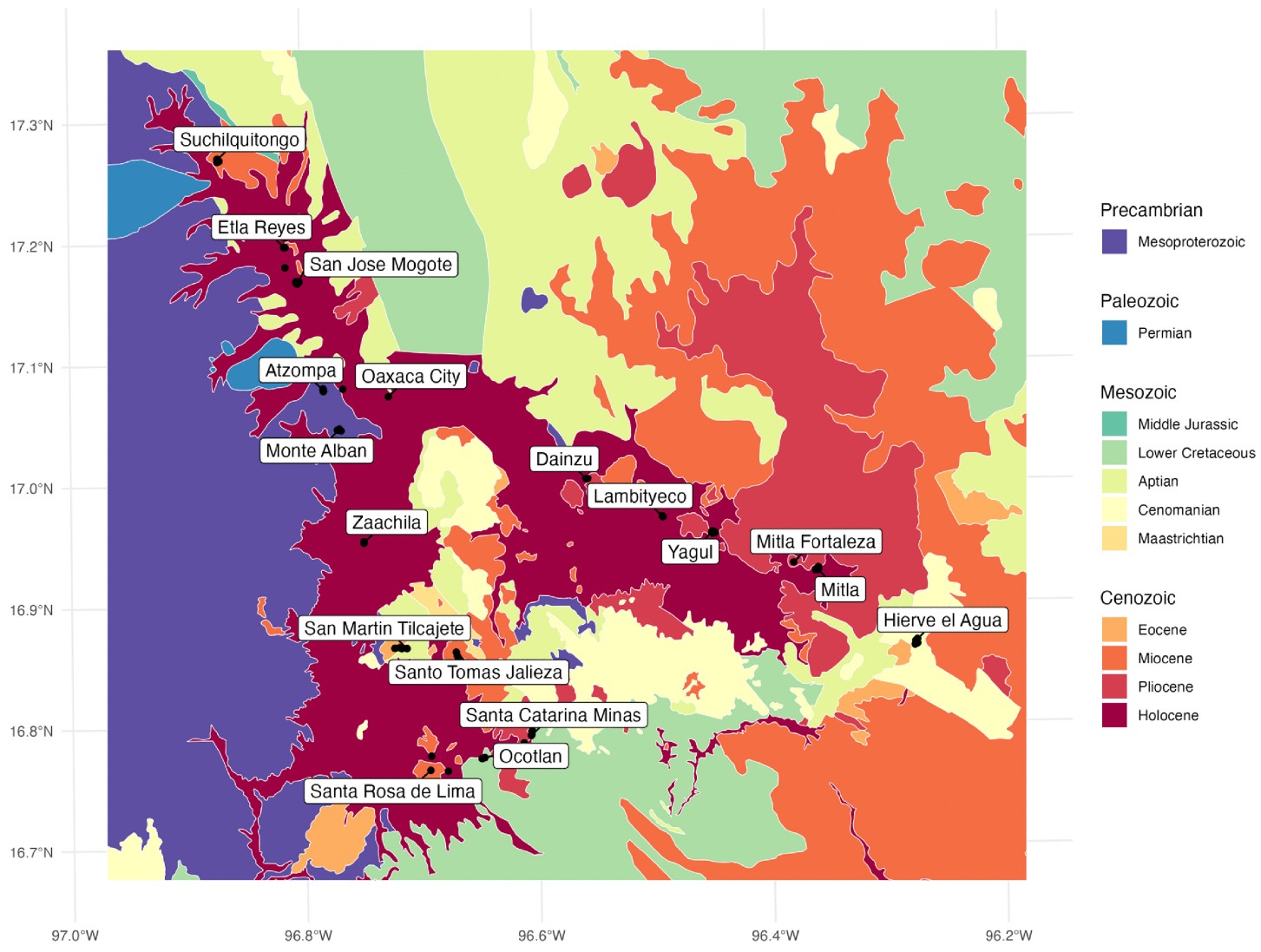

**Fig 2. A map of the geologic age of the underlying bedrock in the Valley of Oaxaca.** Map created with publicly available state boundaries from Natural Earth [86]. Geologic bedrock age data adapted from the Mexican Geological Service [87].

bioavailable strontium with non-local sources of strontium. Collected samples primarily constituted plant leaves or seed pods, not whole plants. We recorded Universal Transverse Mercator (UTM) coordinate and elevation data for each plant sample using a hand-held GPS unit.

No IRB approval or waiver was sought for this research, as it does not involve human subjects. Field research was conducted under the Mexican Instituto Nacional de Antropología e Historia (INAH)'s Consejo de Arqueología Oficio 401.B(4)19.2016/36/1020 and 401.B(4)19.2016/36/1065. INAH does not require specific permissions to collect modern plant samples from the study sites. Furthermore, no endangered or protected plant species were collected for the study. Collected plant samples were imported for analysis to the Arizona State University Archaeological Chemistry Laboratory under permits granted to Pacheco-Forés from the United States Department of Agriculture Animal and Plant Health Inspection Service (permits PCIP-17–00469 and PCIP-18–00287). Additional information regarding the ethical, cultural, and scientific considerations specific to inclusivity in global research is included in the Supporting Information (S1 Checklist).

## Biogeochemical methods

Plant samples were prepared at the Arizona State University Archaeological Chemistry Laboratory. When possible, pre-hispanic diets were simulated through the manual isolation and analysis of edible components of dried plants [91]. Plant samples were prepared rinsed with 18.2 MΩ Millipore water to remove adhering dirt and ashed in a furnace for 10 hours at 800° C. Approximately 25 mg of plant ash was digested in 2 mL of concentrated nitric and hydrochloric acid (one part $HNO_3$ to three parts HCl) at 50° C for 24 hours. This aggressive leach does not include hydrofluoric acid, which would break down the silica tetrahedra structure of most silicate minerals. The leach leaves much of the soil in a solid form while prioritizing the release of bioavailable strontium within plants. The supernatant leach solution was evaporated, and sample precipitates were redissolved in concentrated nitric acid and diluted to a 2M stock solution.

Dissolved samples were analyzed at the Metals, Environmental, and Terrestrial Analytical Laboratory (METAL) at Arizona State University. An aliquot was taken for elemental concentration analysis on a Thermo Fisher Scientific iCAP quadrupole inductively coupled plasma mass spectrometer (Q-ICP-MS). Strontium was then separated with a Prep*FAST* system following the standard laboratory protocol as reported in [92]. Strontium was isolated from the sample matrix using Elemental Scientific, Inc. supplied Sr-Ca ion exchange resin (Part CF-MC-SrCa-1000) and ultrapure 5M nitric acid ($HNO_3$). To determine chemical recovery, 22% of samples were measured on the Q-ICP-MS before and after chemical purification. Average chemistry yield was 85%.

The remaining portion of each Sr cut from the Prep*FAST* was dried down in a Teflon beaker and digested with concentrated $HNO_3$ and 30% hydrogen peroxide ($H_2O_2$) to remove organics from the resin. Once digested, samples were again dried down and reconstituted with 0.32 M $HNO_3$. Using concentration information from the Q-ICP-MS, the samples were diluted with 0.32 M $HNO_3$ to a calculated constant concentration of 50 ppb Sr. Samples were then run on a Thermo-Finnigan Neptune multi-collector inductively coupled plasma mass spectrometer (MC-ICP-MS).

Data were collected by measuring 60 simultaneous ratios integrating 4.194 seconds each. Samples were corrected for on-peak blanks, and in-line correction of the contributions of $^{84}Kr$ on $^{84}Sr$ and $^{86}Kr$ on $^{86}Sr$ using $^{83}Kr/^{84}Kr$ ratio of 0.201750 and $^{83}Kr/^{86}Kr$ ratio of 0.664533, after instrumental mass bias correction using a normalizing $^{88}Sr/^{86}Sr$ ratio of 8.375209. Samples were analyzed in two different analytical sessions. Typical sensitivity was 36 V on $^{88}Sr$ with a 50 ppb Sr solution, with $^{83}Kr$ values of 0.0001 V. $^{85}Rb$ voltages for samples were typically <0.001 V due to the low Rb/Sr initial ratios of the samples and effective chemical purification, but all data was interference-corrected using an $^{85}Rb/^{87}Rb$ ratio of 2.588960, normalized to $^{88}Sr/^{86}Sr$ as above. Ratio outliers two standard deviations outside the mean were removed using a Matlab 2D-mathematical correction routine written by Stephen Romaniello. All data were renormalized to a value of SRM 987 of 0.71025 using bracketing standards to compensate for instrument and Kr background drift. Typical internal $^{87}Sr/^{86}Sr$ two standard error (SE) precision was ≤ 1e-5.

Sequences included bracketing concentration-matched SRM 987 standards. SRM 987 was run as a bracketing standard with a measured value of $^{87}Sr/^{86}Sr = 0.710256 \pm 0.000010$ (2σ, $n = 57$). Each analytical session included a sequence incorporating SRM 987 standard in a range of variable concentrations to verify the accuracy of $^{87}Sr/^{86}Sr$ values for samples; reported values are all fall within the range for accurate $^{87}Sr/^{86}Sr$ values within the range of error of the bracketing standards [93]. In addition, SRM 987 doped with calcium up to a ratio of Ca/Sr of 500 was run to simulate the accuracy and precision of isotope ratios in poorly purified samples with low yields. This solution of SRM 987 run at 50% concentration doped to a Ca/Sr of 500 was run as a check standard with a measured value of $^{87}Sr/^{86}Sr = 0.710252 \pm 0.000018$ (2σ, $n = 17$). IAPSO seawater (Ocean Scientific International Ltd., Havant, UK) as a secondary check standard had a measured value of $0.709182 \pm 0.000009$ (2σ, $n = 13$), within error of the published value of $0.709182 \pm 0.000004$ [94]. NIST 1400 purified in parallel with samples had a measured value of $0.713114 \pm 0.000013$ (2σ, $n = 10$), within error of the published value of $0.713150 \pm 0.000016$ [95]. Approximately 5% of samples ($n = 7$) were chemically separated in triplicate, with an average precision on measurements of $\pm 0.000009$ (2σ).

## Isoscape model development and testing

We compiled strontium isotope data for North and South America from existing global and regional syntheses [8,67,72] and our newly collected measurements from the Valley of Oaxaca. The compiled dataset included 7,271 measurements from humans, animals, plants, soil, and water sources across diverse environments. Human data from the Caribbean and Mesoamerica Biogeochemical Isotope Overview (CAMBIO, version 1.2) were filtered to exclude individuals identified as non-local from the compiled dataset [8]. Water samples present a special challenge for strontium modeling, as they can reflect non-local signals due to transport across watersheds [69,96–98]. Rather than excluding these samples entirely, we incorporated a categorical predictor variable indicating sample type (e.g., water, plant, etc.), allowing the model to appropriately handle the different uncertainty profiles associated with each sample type.

We filtered the data to exclude samples with $^{87}Sr/^{86}Sr$ values outside the range of 0.703 to 0.780 to remove potential outliers and focus model training on the most relevant range for archaeological interpretation within Mesoamerica [see 8]. We applied a logit transformation to the strontium isotope ratios, a standard approach for bounded variables that helps normalize the distribution and reduces the influence of extreme values on model training.

We assembled a set of spatial environmental predictors based on established relationships with strontium isotope distribution patterns [67,69]. These predictors included updated climate variables from CHELSA V2.1 [99,100], soil properties at multiple depths such as bulk density and pH [101], geological variables including basement age and lithological classifications [67,102,103], aerosol deposition data from MERRA-2 reanalysis [104], and terrain variables. We performed domain-specific principal component analyses to reduce the dimensionality of these predictors, creating a minimal set of uncorrelated climate, soil, geological, and aerosol variables. This approach resulted in 24 predictor variables for the final model. Detailed PCA results are provided in S1 Fig.

We employed a Bayesian Additive Regression Trees (BART) approach [105] for modeling strontium isoscapes. Traditional random forests used in previous isoscapes [e.g., 67], have limitations in quantifying prediction uncertainty, while quantile regression forests [e.g., 63,75,76] underestimate uncertainty in regions with sparse training data. BART addresses these limitations by extending ensemble decision tree methods with Bayesian statistical approaches. Unlike traditional random forests that use fixed tree structures, BART treats the tree configurations themselves as uncertain, maintaining probability distributions over different possible tree arrangements. This Bayesian framework allows BART to explicitly model both natural variability in isotopic ratios and uncertainty arising from limited model knowledge, while preventing overfitting through regularization.

The key advantage of BART for archaeological applications is its ability to naturally quantify uncertainty that appropriately increases in data-sparse or geologically complex regions. Rather than providing only a single predicted value for each location, BART generates a range of plausible values with associated probabilities. This approach gives archaeologists crucial information about the relative confidence of local strontium ranges and by extension the reliability of migrant identification. In regions with higher prediction uncertainty, wider ranges must be considered for defining "local" signatures, while areas with lower uncertainty allow for more precise determinations.

We evaluated model performance using both root mean square error (RMSE) and R-squared ($R^2$) values on held-out test data. RMSE represents the average prediction error in the same units as the data ($^{87}Sr/^{86}Sr$), while $R^2$ represents the proportion of variance in the observed data explained by the model. A lower RMSE and a higher $R^2$ indicate better model performance. Our iterative evaluation approach involved first training a model on our compiled dataset from North and South America, excluding all data (both compiled and observed) from Mesoamerica. We then tested this model's prediction accuracy on compiled data from Mesoamerica, using this process to fine tune the model's parameters while assessing the model's ability to extrapolate to a new region of interest. Then we retrained this model with the compiled data from Mesoamerica and tested it on the newly collected Oaxacan samples to assess its ability to extrapolate within our region of interest. For the final isoscape, we retrained the model incorporating all compiled data from North, Central,

and South America and our new Valley of Oaxaca samples. We compare the model predictions and performance metrics at each step in the process to assess the stability of the model's predictions and the added value of including increasingly local samples in the training set. We then applied this final model across the Valley of Oaxaca study area to create spatially explicit predictions of $^{87}Sr/^{86}Sr$ values with associated uncertainty estimates. We assessed variable importance by calculating the frequency with which each variable was used in the BART model [105], allowing us to identify the major environmental factors driving strontium isotope patterns in the region (S1 Fig). The resulting isoscape provides both predicted $^{87}Sr/^{86}Sr$ values and associated uncertainty estimates at a 1 km spatial resolution. These predictions serve as a baseline against which archaeological samples can be compared to distinguish local from non-local individuals in the region.

## Results

### Biogeochemical results

Table 2 reports observed $^{87}Sr/^{86}Sr$ values in sampled plants. In the Valley of Oaxaca, $^{87}Sr/^{86}Sr$ values ranged from 0.704752 to 0.711976, with a mean of $^{87}Sr/^{86}Sr = 0.706851 \pm 0.001419$ ($1\sigma$, $n = 95$).

### Isoscape model performance

The continental-only strontium model, trained on compiled data from North and South America but excluding Mesoamerica, achieved moderate predictive performance when applied to Mesoamerican test data (logit-transformed RMSE = 0.42; raw $R^2 = 0.39$). Incorporating Mesoamerican samples into the training data substantially improved in-sample fit within the region (logit RMSE = 0.18; raw RMSE = 0.0011; raw $R^2 = 0.71$), highlighting the value of regional calibration.

When tested on the independent Oaxaca samples not included in the model training, the continental-only model showed limited but nontrivial explanatory power (logit RMSE = 0.34; raw RSME = 0.0011; raw $R^2 = 0.33$). Adding Mesoamerican data to training improved extrapolation to Oaxaca, lowering error to 0.00087 in raw units (logit RMSE = 0.26; raw $R^2 = 0.30$).

In comparison, the geological bedrock-only benchmark performed worse in both regions, with raw RMSE = 0.0019 and $R^2 = 0.24$ for Mesoamerica, and raw RMSE = 0.0027 and $R^2 = 0.37$ for Oaxaca. These contrasts demonstrate that machine-learning isoscapes, even when trained at continental scale, provide clear gains over geological baselines, and that incorporating regional samples further enhances predictive accuracy without sacrificing extrapolation capacity (Table 3). In practical terms, out-of-sample generalization to Oaxaca achieved errors of less than one part in a thousand of the $^{87}Sr/^{86}Sr$ ratio, underscoring the utility of regionally calibrated models. For the subsequent spatial predictions, however, we fit the final model to the full continental and Mesoamerican dataset (including Oaxaca) to maximize coverage and resolution.

### Predictive variables

Variable importance analysis revealed several key factors driving strontium isotope distribution patterns in the region. The most influential predictors were distance from the coast; basement age of underlying geology; the first principal component of the strontium bedrock model; climate principal components related to precipitation intensity and timing; aerosol variables related to carbon and sulfate abundance and dust concentration; and soil variables related to cation exchange capacity and texture (S1 Fig). These findings align with theoretical expectations for strontium cycling, where the basic bedrock geology is filtered through soil transport processes, climatic weathering, and aerosol deposition [67]. Notably, the high-resolution distance from the coast was favored over the low-resolution sea salt aerosol map in the model, suggesting the clear added value of high-resolution predictors, even if they are indirect proxies for "true" driving variables, at estimating local Sr baselines.

**Table 2.** Provenance data, sample details, and $^{87}$Sr/$^{86}$Sr values for analyzed Valley of Oaxaca plant samples.

| Laboratory ID | Site | UTM-E[a] | UTM-N[a] | Elevation (masl) | Species | Plant Rooting Depth | $^{87}$Sr/$^{86}$Sr | 2 SE |
|---|---|---|---|---|---|---|---|---|
| ACL-10588 | Atzompa | 736019 | 1889532 | 1817 | *Agave* spp. | shallow | 0.707597 | 0.000008 |
| ACL-10589 | Atzompa | 736018 | 1889624 | 1983 | *Yucca filifera* | deep | 0.707703 | 0.000008 |
| ACL-10590 | Atzompa | 737791 | 1889737 | 1884 | *Jacaranda mimosifolia* | shallow | 0.707704 | 0.000012 |
| ACL-10591 | Atzompa | 735986 | 1889778 | 1893 | *Acacia farnesiana* | deep | 0.707504 | 0.000007 |
| ACL-10592 | Atzompa | 735987 | 1889716 | 1884 | *Opuntia ficus* | shallow | 0.707301 | 0.000011 |
| ACL-10593 | Atzompa | 735988 | 1889655 | 1710 | *Jacaranda mimosifolia* | shallow | 0.707822 | 0.000009 |
| ACL-9642 | Dainzú | 760073 | 1881555 | 1617 | *Poa* spp. | shallow | 0.705759 | 0.000008 |
| ACL-9643 | Dainzú | 760018 | 1881603 | 1615 | *Opuntia ficus* | shallow | 0.705911 | 0.000007 |
| ACL-9644 | Dainzú | 759992 | 1881588 | 1611 | *Prosopis* spp. | deep | 0.705724 | 0.000012 |
| ACL-9645 | Dainzú | 760168 | 1881588 | 1637 | *Agave* spp. | shallow | 0.705270 | 0.000009 |
| ACL-10598 | Etla Reyes | 732518 | 1903608 | 1640 | *Opuntia ficus* | shallow | 0.706611 | 0.000008 |
| ACL-10599 | Etla Reyes | 732509 | 1904407 | 1640 | *Prosopis* spp. | deep | 0.706530 | 0.000006 |
| ACL-10600 | Etla Reyes | 732528 | 1902685 | 1640 | *Prosopis* spp. | deep | 0.706536 | 0.000013 |
| ACL-10601 | Etla Reyes | 732410 | 1902715 | 1643 | *Opuntia ficus* | shallow | 0.706549 | 0.000008 |
| ACL-10602 | Etla Reyes | 732410 | 1902684 | 1643 | *Opuntia ficus* | shallow | 0.706487 | 0.000008 |
| ACL-10603 | Etla Reyes | 732520 | 1900840 | 1640 | *Agave* spp. | shallow | 0.706546 | 0.000009 |
| ACL-9629 | Hierve el Agua | 790184 | 1866831 | 1771 | *Agave* spp. | shallow | 0.707431 | 0.000013 |
| ACL-9630 | Hierve el Agua | 790123 | 1866492 | 1711 | *Agave* spp. | shallow | 0.707377 | 0.000008 |
| ACL-9631 | Hierve el Agua | 790343 | 1866610 | 1653 | *Agave* spp. | shallow | 0.707448 | 0.000011 |
| ACL-9632 | Hierve el Agua | 790333 | 1866995 | 1764 | *Agave* spp. | shallow | 0.707428 | 0.000010 |
| ACL-9639 | Lambityeco | 767086 | 1878105 | 1613 | *Yucca filifera* | deep | 0.705545 | 0.000013 |
| ACL-9640 | Lambityeco | 767040 | 1878109 | 1609 | *Agave* spp. | shallow | 0.705730 | 0.000009 |
| ACL-9641 | Lambityeco | 767017 | 1878173 | 1604 | *Prosopis* spp. | deep | 0.705515 | 0.000010 |
| ACL-9633 | Mitla | 781307 | 1873516 | 1694 | *Agave* spp. | shallow | 0.705960 | 0.000010 |
| ACL-9634 | Mitla | 781270 | 1873523 | 1703 | *Poa* spp. | shallow | 0.706405 | 0.000008 |
| ACL-9635 | Mitla | 781208 | 1873324 | 1689 | *Yucca filifera* | deep | 0.705913 | 0.000009 |
| ACL-9636 | Mitla | 781289 | 1873332 | 1691 | *Opuntia ficus* | shallow | 0.705914 | 0.000008 |
| ACL-9637 | Mitla | 781204 | 1873272 | 1695 | *Pachycereus weberi* | shallow | 0.706094 | 0.000006 |
| ACL-9638 | Mitla | 781032 | 1873257 | 1686 | *Poa* spp. | shallow | 0.706143 | 0.000010 |
| ACL-9095 | Mitla Fortaleza | 779009 | 1873969 | 1832 | *Yucca filifera* | deep | 0.705918 | 0.000006 |
| ACL-9096 | Mitla Fortaleza | 779009 | 1873969 | 1832 | *Stenocereus thurberi* | shallow | 0.705937 | 0.000006 |
| ACL-9097 | Mitla Fortaleza | 779009 | 1873969 | 1832 | *Agave* spp. | shallow | 0.705900 | 0.000011 |
| ACL-9098 | Mitla Fortaleza | 779009 | 1873969 | 1832 | *Agave* spp. | shallow | 0.705907 | 0.000014 |
| ACL-9099 | Mitla Fortaleza | 779009 | 1873969 | 1832 | *Agave* spp. | shallow | 0.706536 | 0.000009 |
| ACL-9100 | Mitla Fortaleza | 779009 | 1873969 | 1832 | *Agave* spp. | shallow | 0.705906 | 0.000007 |
| ACL-9091 | Monte Albán | 737631 | 1885941 | 1940 | *Agave* spp. | shallow | 0.708005 | 0.000007 |
| ACL-9092 | Monte Albán | 737631 | 1885941 | 1940 | *Agave* spp. | shallow | 0.707878 | 0.000007 |
| ACL-9093 | Monte Albán | 737631 | 1885941 | 1940 | *Leucaena leucocephala* | deep | 0.707691 | 0.000008 |
| ACL-9094 | Monte Albán | 737631 | 1885941 | 1940 | *Opuntia ficus* | shallow | 0.707726 | 0.000014 |
| ACL-10594 | Monte Albán | 737301 | 1886010 | 1881 | *Agave* spp. | shallow | 0.707935 | 0.000010 |
| ACL-10595 | Monte Albán | 737449 | 1886012 | 1929 | *Prosopis* spp. | deep | 0.707902 | 0.000007 |
| ACL-10596 | Monte Albán | 737448 | 1886073 | 1929 | *Agave* spp. | shallow | 0.707607 | 0.000008 |
| ACL-10597 | Monte Albán | 737478 | 1886104 | 1926 | *Jacaranda mimosifolia* | shallow | 0.707732 | 0.000006 |
| ACL-10618 | Ocotlán | 755149 | 1856142 | 1475 | *Pachycereus weberi* | shallow | 0.705086 | 0.000009 |
| ACL-10619 | Ocotlán | 750794 | 1856090 | 1475 | *Agave* spp. | shallow | 0.705144 | 0.000008 |
| ACL-10620 | Ocotlán | 750558 | 1855995 | 1570 | *Acacia farnesiana* | deep | 0.705119 | 0.000008 |

*(Continued)*

**Table 2.** (Continued)

| Laboratory ID | Site | UTM-E[a] | UTM-N[a] | Elevation (masl) | Species | Plant Rooting Depth | $^{87}$Sr/$^{86}$Sr | 2 SE |
|---|---|---|---|---|---|---|---|---|
| ACL-9611 | San José Mogote | 733521 | 1899560 | 1615 | *Prosopis* spp. | deep | 0.706327 | 0.000008 |
| ACL-9612 | San José Mogote | 733501 | 1899500 | 1616 | *Opuntia ficus* | shallow | 0.706104 | 0.000007 |
| ACL-9613 | San José Mogote | 733643 | 1899366 | 1617 | *Agave* spp. | shallow | 0.706062 | 0.000006 |
| ACL-9614 | San José Mogote | 733622 | 1899359 | 1606 | *Prosopis* spp. | deep | 0.706574 | 0.000008 |
| ACL-9615 | San José Mogote | 733802 | 1899534 | 1608 | *Opuntia ficus* | shallow | 0.706260 | 0.000012 |
| ACL-9616 | San José Mogote | 733748 | 1899569 | 1612 | *Prosopis* spp. | deep | 0.706270 | 0.000008 |
| ACL-9083 | San Martín Tilcajete | 745202 | 1867422 | 1623 | *Acacia farnesiana* | deep | 0.708259[b] | 0.000011 |
| ACL-9084 | San Martín Tilcajete | 745202 | 1867422 | 1623 | *Agave* spp. | shallow | 0.709700[b] | 0.000013 |
| ACL-9085 | San Martín Tilcajete | 745202 | 1867422 | 1623 | *Opuntia ficus* | shallow | 0.709633[b] | 0.000010 |
| ACL-9086 | San Martín Tilcajete | 745202 | 1867422 | 1623 | *Acacia farnesiana* | deep | 0.710458[b] | 0.000010 |
| ACL-9087 | San Martín Tilcajete | 745202 | 1867422 | 1623 | *Yucca filifera* | deep | 0.707351[b] | 0.000008 |
| ACL-9088 | San Martín Tilcajete | 745202 | 1867422 | 1623 | *Opuntia ficus* | shallow | 0.707237[b] | 0.000006 |
| ACL-9089 | San Martín Tilcajete | 745202 | 1867422 | 1623 | *Acacia farnesiana* | deep | 0.708389[b] | 0.000009 |
| ACL-9090 | San Martín Tilcajete | 745202 | 1867422 | 1623 | *Dichondra argentea* | shallow | 0.708531[b] | 0.000006 |
| ACL-10604 | San Martín Tilcajete | 743657 | 1866063 | 1545 | *Agave* spp. | shallow | 0.708553[b] | 0.000010 |
| ACL-10605 | San Martín Tilcajete | 743183 | 1866057 | 1579 | *Ipomoea arborescens* | shallow | 0.711976[b] | 0.000010 |
| ACL-10606 | San Martín Tilcajete | 743005 | 1866148 | 1561 | *Protium copal* | deep | 0.711739[b] | 0.000007 |
| ACL-10607 | San Martín Tilcajete | 742561 | 1866081 | 1612 | *Pachycereus weberi* | shallow | 0.709570[b] | 0.000006 |
| ACL-10614 | Santa Catarina Minas | 754393 | 1857425 | 1612 | *Acacia farnesiana* | deep | 0.705971 | 0.000009 |
| ACL-10615 | Santa Catarina Minas | 755036 | 1858109 | 1615 | *Agave* spp. | shallow | 0.705627 | 0.000008 |
| ACL-10616 | Santa Catarina Minas | 755065 | 1858202 | 1631 | *Opuntia ficus* | shallow | 0.705870 | 0.000009 |
| ACL-10617 | Santa Catarina Minas | 755120 | 1858541 | 1612 | *Pachycereus weberi* | shallow | 0.706535 | 0.000006 |
| ACL-10621 | Santa Rosa de Lima | 745904 | 1856217 | 1527 | *Agave* spp. | shallow | 0.705435 | 0.000015 |
| ACL-10623 | Santa Rosa de Lima | 745830 | 1854925 | 1533 | *Jacaranda mimosifolia* | shallow | 0.705533 | 0.000008 |
| ACL-10624 | Santa Rosa de Lima | 747431 | 1854851 | 1527 | *Pachycereus weberi* | shallow | 0.705443 | 0.000009 |
| ACL-10608 | Santo Tomás Jalieza | 748316 | 1865287 | 1609 | *Agave* spp. | shallow | 0.705024 | 0.000007 |
| ACL-10609 | Santo Tomás Jalieza | 748255 | 1865409 | 1585 | *Ipomoea arborescens* | shallow | 0.704752 | 0.000006 |
| ACL-10610 | Santo Tomás Jalieza | 748193 | 1865655 | 1585 | *Pinus* spp. | deep | 0.704952 | 0.000007 |
| ACL-10611 | Santo Tomás Jalieza | 748136 | 1867930 | 1615 | *Protium copal* | deep | 0.706335 | 0.000009 |
| ACL-10612 | Santo Tomás Jalieza | 748162 | 1865746 | 1597 | *Acacia farnesiana* | deep | 0.704809 | 0.000011 |
| ACL-10613 | Santo Tomás Jalieza | 748162 | 1865716 | 1588 | *Agave* spp. | shallow | 0.704783 | 0.000008 |
| ACL-9617 | Suchilquitongo | 726283 | 1910455 | 1786 | *Agave* spp. | shallow | 0.706545 | 0.000008 |
| ACL-9618 | Suchilquitongo | 726451 | 1910539 | 1804 | *Prosopis* spp. | deep | 0.706795 | 0.000010 |
| ACL-9619 | Suchilquitongo | 726293 | 1910688 | 1810 | *Agave* spp. | shallow | 0.706946 | 0.000015 |
| ACL-9620 | Suchilquitongo | 726292 | 1910717 | 1808 | *Opuntia ficus* | shallow | 0.706938 | 0.000005 |
| ACL-9621 | Suchilquitongo | 726289 | 1910739 | 1809 | *Agave* spp. | shallow | 0.707002 | 0.000013 |
| ACL-9622 | Suchilquitongo | 726309 | 1910503 | 1815 | *Agave* spp. | shallow | 0.706519 | 0.000008 |
| ACL-9646 | Yagul | 771545 | 1876618 | 1683 | *Opuntia ficus* | shallow | 0.705875 | 0.000011 |
| ACL-9647 | Yagul | 771529 | 1876694 | 1690 | *Opuntia ficus* | shallow | 0.705920 | 0.000006 |
| ACL-9648 | Yagul | 771582 | 1876791 | 1700 | *Pachycereus weberi* | shallow | 0.706043 | 0.000010 |
| ACL-9649 | Yagul | 771649 | 1876788 | 1699 | *Opuntia ficus* | shallow | 0.705815 | 0.000010 |
| ACL-9650 | Yagul | 771826 | 1876657 | 1727 | *Opuntia ficus* | shallow | 0.705874 | 0.000009 |
| ACL-9651 | Yagul | 771800 | 1876708 | 1740 | *Agave* spp. | shallow | 0.705855 | 0.000011 |
| ACL-9623 | Zaachila | 739726 | 1875665 | 1525 | *Prosopis* spp. | deep | 0.708502 | 0.000005 |
| ACL-9624 | Zaachila | 739719 | 1875686 | 1535 | *Prosopis* spp. | deep | 0.708782 | 0.000009 |

*(Continued)*

**Table 2.** (Continued)

| Laboratory ID | Site | UTM-E[a] | UTM-N[a] | Elevation (masl) | Species | Plant Rooting Depth | $^{87}$Sr/$^{86}$Sr | 2 SE |
|---|---|---|---|---|---|---|---|---|
| ACL-9625 | Zaachila | 739735 | 1875711 | 1532 | *Agave* spp. | shallow | 0.708395 | 0.000007 |
| ACL-9626 | Zaachila | 739722 | 1875763 | 1529 | *Agave* spp. | shallow | 0.707974 | 0.000008 |
| ACL-9627 | Zaachila | 739718 | 1875801 | 1514 | *Opuntia ficus* | shallow | 0.708248 | 0.000014 |
| ACL-9628 | Zaachila | 739699 | 1875702 | 1526 | *Agave* spp. | shallow | 0.708384 | 0.000008 |

[a]All UTM coordinates are in UTM Zone 14N.

[b]We identified these $^{87}$Sr/$^{86}$Sr values as implausibly high for the region and likely the artifact of the use of agricultural fertilizers in the collection area. See further discussion below and in S1 Appendix. These samples were excluded from the final model training.

**Table 3. Model performance metrics[a] across different training datasets and test regions.**

| Training Data | Test Data | Logit RMSE | Logit R² | Raw RMSE | Raw R² |
|---|---|---|---|---|---|
| **Out-of-sample tests** | | | | | |
| Continental (no Meso) | Mesoamerica[b] | 0.42 | 0.60 | 0.0017 | 0.39 |
| Continental (no Meso) | Oaxaca | 0.34 | 0.35 | 0.0011 | 0.33 |
| Continental + Meso (no Oaxaca) | Oaxaca | 0.26 | 0.29 | 0.00087 | 0.30 |
| Bedrock baseline[c] | Mesoamerica[b] | 0.54 | 0.39 | 0.0019 | 0.24 |
| Bedrock baseline[c] | Oaxaca | 0.55 | 0.36 | 0.0027 | 0.37 |
| **In-sample fits[d]** | | | | | |
| Continental + Meso (no Oaxaca) | Mesoamerica[b] | 0.18 | 0.89 | 0.0011 | 0.71 |
| Continental + Meso + Oaxaca | Mesoamerica[b] | 0.18 | 0.89 | 0.0011 | 0.71 |
| Continental + Meso + Oaxaca | Oaxaca | 0.22 | 0.5 | 0.0007 | 0.49 |

[a]Values show RMSE (Root Mean Square Error) and R² (coefficient determination) in both logit-transformed and untransformed (original) $^{87}$Sr/$^{86}$Sr units. Lower RMSE and higher R² values indicate better model performance.

[b]Mesoamerica is defined as any $^{87}$Sr/$^{86}$Sr values from Mexico, Belize, Guatemala, Honduras, El Salvador, Nicaragua, and Costa Rica.

[c]The bedrock-only benchmark represents a geologically-informed baseline [from 67] for comparison with the machine learning models.

[d]Indicates measures of in-sample performance which may be over optimistic.

## Spatial distribution of predicted values

The final calibrated model produced estimates of $^{87}$Sr/$^{86}$Sr values across Mesoamerica ranging from 0.704 to 0.712 (Fig 3; upper and lower 95% confidence intervals for $^{87}$Sr/$^{86}$Sr prediction ranges are available in S2 Fig). The model revealed several distinct isotopic regions. Areas with elevated $^{87}$Sr/$^{86}$Sr values (>0.709) were identified in the Maya Mountains, Yucatán Peninsula, and coastal ranges of Oaxaca, consistent with older geological formations in these regions and available published $^{87}$Sr/$^{86}$Sr data [106]. Less radiogenic areas (<0.706) were observed across the Tuxtla Mountains in Veracruz and along a strip of the central volcanic axis, reflecting younger volcanic geology. Interestingly, the fine-tuned model incorporating Oaxaca plant samples reduced the estimated strontium isotope ratios across most of Mexico, only raising estimates slightly in the southern coastal regions of Oaxaca and Guerrero. On the whole, isoscape predicted values and ranges generally conform to published regional summary $^{87}$Sr/$^{86}$Sr data across Mesoamerica [55] while providing a clearer picture of regional strontium spatial variability.

In the Valley of Oaxaca, our isoscape suggests that we should be able to detect the intravalley mobility suggested by archaeological settlement survey data (Fig 4). Although each arm of the Valley exhibits some overlap in predicted $^{87}$Sr/$^{86}$Sr values, the eastern Tlacolula arm of the Valley is dominated by large areas of lower predicted $^{87}$Sr/$^{86}$Sr values (~0.705–0.706) that conform with expectations from its younger Mesozoic and Cenozoic underlying geology. In contrast, the northwestern Etla arm has higher predicted $^{87}$Sr/$^{86}$Sr values (~0.706–0.707), and the southern Ocotlán-Zimatlán arm

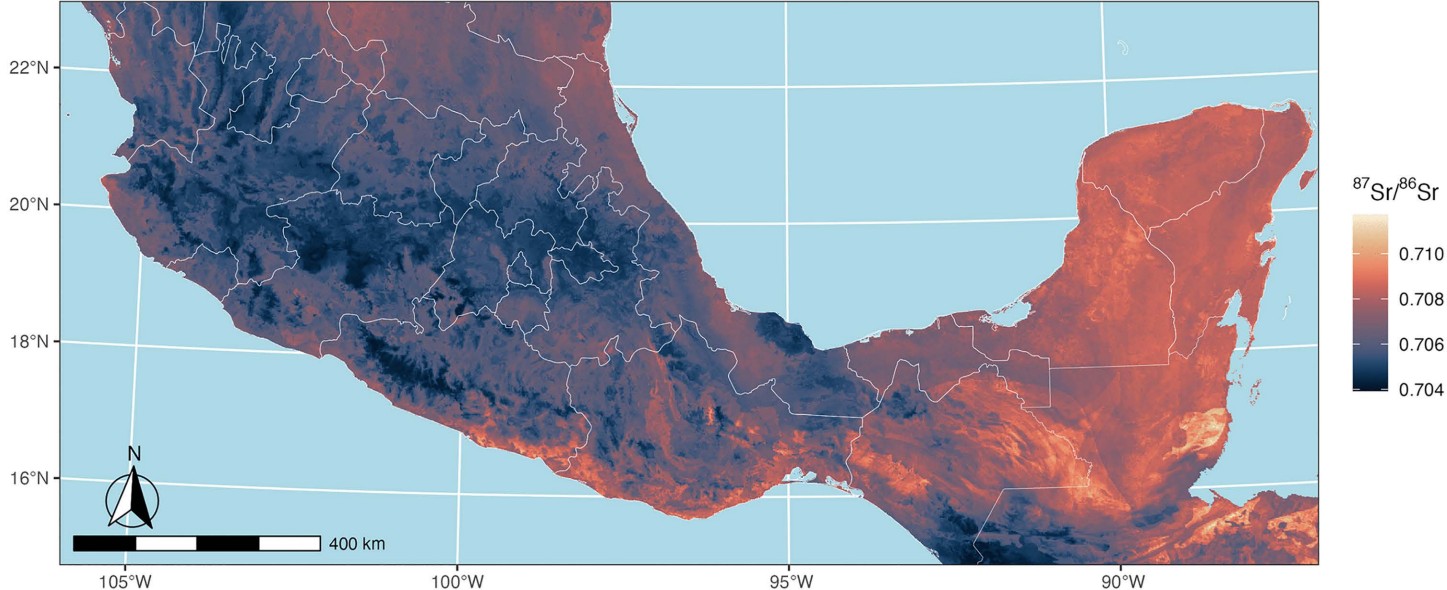

**Fig 3. Final calibrated isoscape model of $^{87}$Sr/$^{86}$Sr variation across Mesoamerica.** Map created with publicly available country and Mexican state boundaries from Natural Earth [86].

features some of the highest predicted $^{87}$Sr/$^{86}$Sr values within the Valley (~0.7075–0.708). These higher values in the Etla and Octolán-Zimatlán arms conform with geological expectations based on their proximity to the older Precambrian Oaxacan Complex. Thus, while the isoscape certainly allows for the identification of non-local individuals within the Valley of Oaxaca who may have originated from other parts of the Valley, the lack of mutually exclusive $^{87}$Sr/$^{86}$Sr values between the arms limits our ability to geolocate from where exactly these non-locals originated.

## Uncertainty patterns

The BART modeling approach provided native uncertainty estimates for all predictions across the isoscape (Fig 5). Consistent with previous studies, we found that strontium prediction uncertainty increased with the strontium ratio, with higher, more radiogenic ratios being associated with greater uncertainty due to increased variability in weathering and transport processes in regions with older, more radiogenic bedrock. Older bedrock has a greater variation in $^{87}$Sr/$^{86}$Sr ratios between coexisting minerals, which have a different susceptibility to weathering. The match between model predictions and strontium observations was generally good, with no clear spatial structure in the residuals that would indicate significant unmodeled processes. However, we did observe lingering mismatches between modeled and observed isotope ratios in specific contexts. These discrepancies primarily occurred in human burial contexts near major bedrock transitions, where relatively high variability in strontium ratios would be expected within the local catchment. Examples include individuals from the Chalillo Reservoir on the Macal River at the edge of the isotopically distinct Maya Mountains, as well as individuals from regional centers like Cholula, a central Mexican pilgrimage site, and Tikal, a dominant Classic period Maya city-state. During our analysis, we also detected a group of samples from our Oaxaca plants at San Martín Tilcajete with implausibly high strontium ratios for the region (Table 2). Further investigation revealed these samples likely reflect contamination from modern agricultural fertilizer application (S1 Appendix) and were excluded from the final model training.

## Discussion

Our results indicate much less clear regional isotopic distinctions that might be suggested by underlying bedrock alone, reflecting the varying influence of climate and soil processes on bioavailable $^{87}$Sr/$^{86}$Sr. The uncertainty in predictions, while

                                                                                          

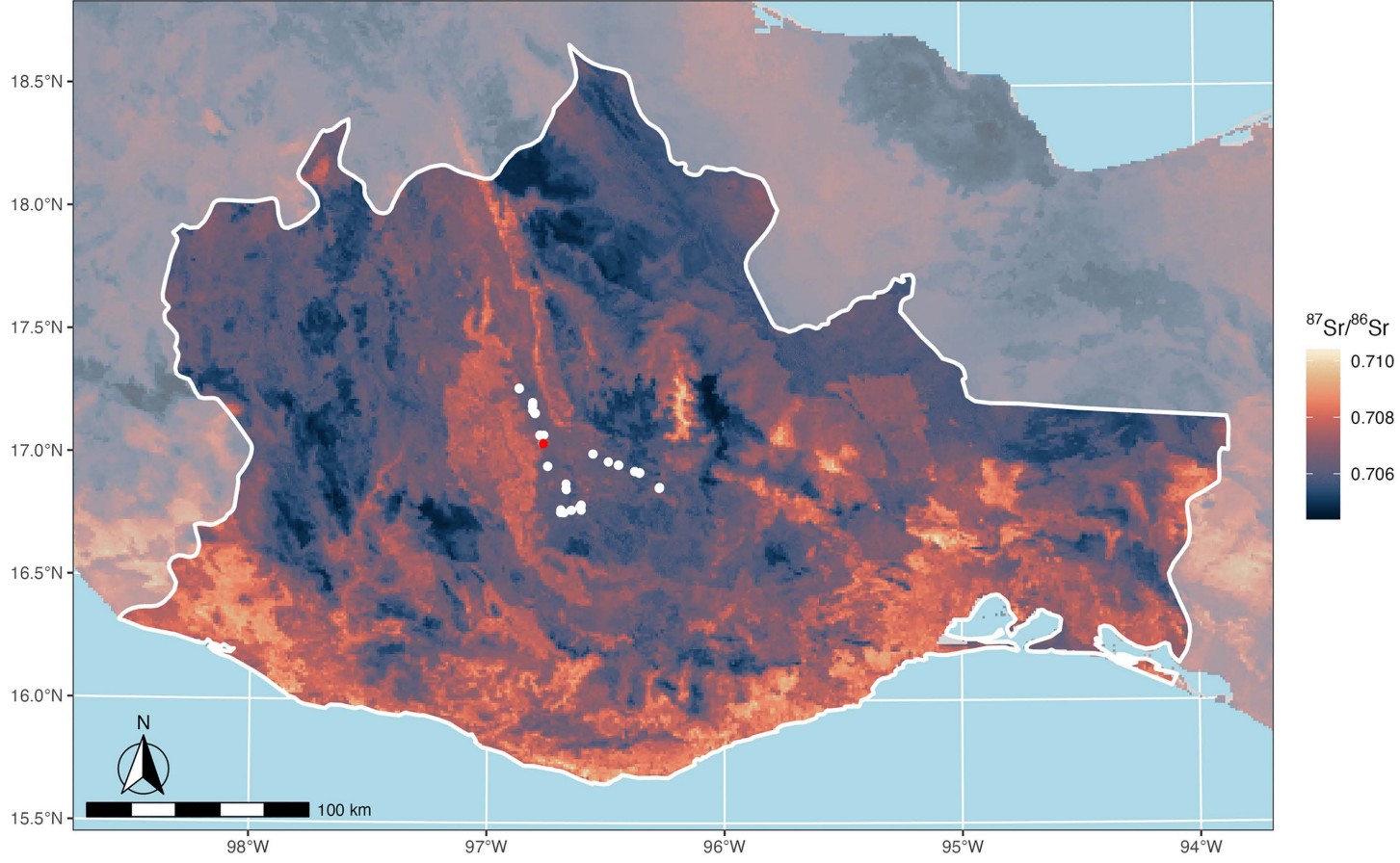

**Fig 4. Final calibrated ⁸⁷Sr/⁸⁶Sr isoscape model across the modern state of Oaxaca.** Sites from which we collected plant samples with which to train the BART model are shown in white, with the exception of Monte Albán, which is shown in red. Map created with publicly available country and Mexican state boundaries from Natural Earth [86].

reduced from the global model, remains relatively high compared to the range of most values. Assessments of migration based on strontium isotopes may thus be more variable than expected from bedrock geology alone.

### Using the predictive ⁸⁷Sr/⁸⁶Sr isoscape

To demonstrate the utility of our predictive ⁸⁷Sr/⁸⁶Sr isoscape model, we applied it to published human ⁸⁷Sr/⁸⁶Sr values (*n* = 10) from Monte Albán [10]. The observations comprise paired ⁸⁷Sr/⁸⁶Sr bone and enamel samples from five individuals dating from the Monte Albán II phase (100 BCE – 200 CE) and the Monte Albán III phase (200–500 CE). Price and colleagues [10] identify one individual's bone and tooth ⁸⁷Sr/⁸⁶Sr values as significantly lower than the other four Monte Albán individuals. The authors suggest this individual likely lived non-locally to Monte Albán during early childhood (enamel) and in the last ~10 years of life (bone). This assessment, however, is not based on an evaluation of biogeochemical baseline ⁸⁷Sr/⁸⁶Sr data, as no previously published environmental ⁸⁷Sr/⁸⁶Sr data exist for Monte Albán.

   Our model's explicit quantification of uncertainty allows for more nuanced interpretation of archaeological samples. For example, when evaluating individuals from Monte Albán, our model provides not just a point estimate for the "local" strontium signature based on the site's 1 km cell within the generated isoscape, but also a probability distribution that accounts for prediction uncertainty (Fig 5). This Bayesian approach offers a more robust framework for identifying potential migrants along a continuous spectrum of locality, as opposed to traditional threshold-based methods that construct a

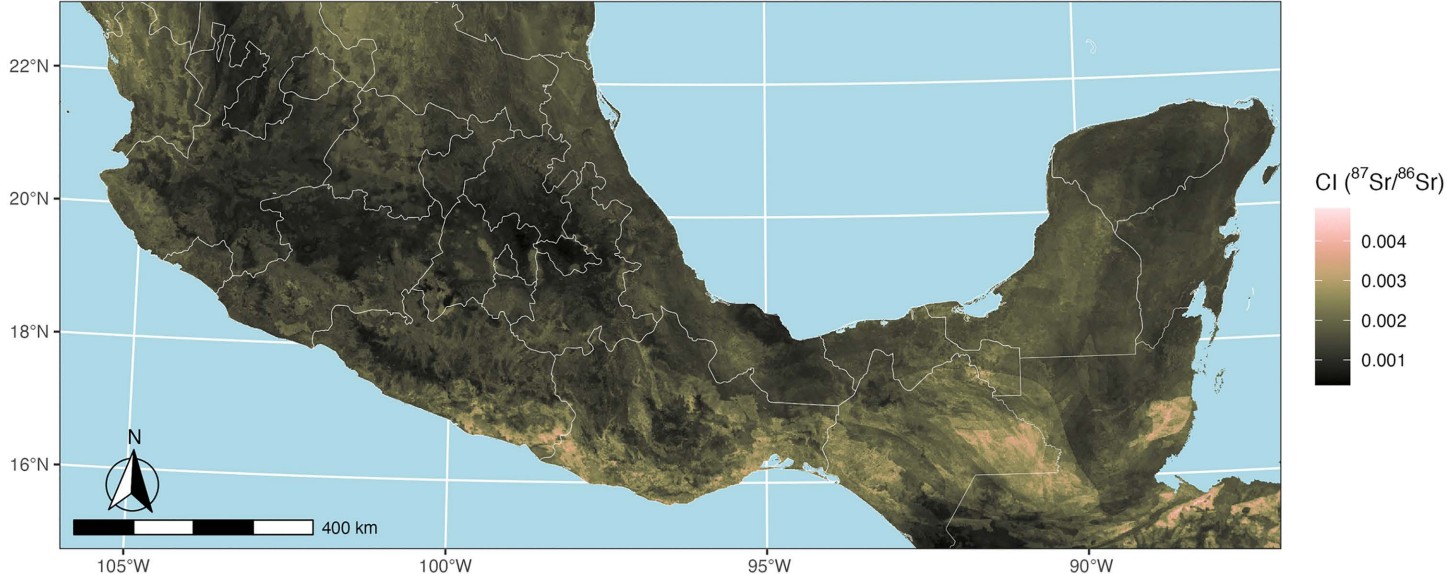

**Fig 5. Native uncertainty estimates for all predicted $^{87}$Sr/$^{86}$Sr values across the isoscape.** Map created with publicly available country and Mexican state boundaries from Natural Earth [86].

rigid binary of local/non-local. Because BART works by growing an ensemble of multiple decision trees, each prediction represents an array of predicted values, one from each tree in the ensemble. This distribution of predicted values makes it straightforward to calculate summary statistics after the fact, such as the mean, median, or upper and lower quantiles of the predicted value at any given pixel.

The isoscape predicts that local Monte Albán $^{87}$Sr/$^{86}$Sr values range from 0.7069 to 0.7086, with a median value of $^{87}$Sr/$^{86}$Sr = 0.7077. This range represents the interquartile range of the final BART model's posterior predictive distribution for $^{87}$Sr/$^{86}$Sr values for Monte Albán (Fig 6) and is somewhat broader than a traditional threshold-based biogeochemical baseline generated from environmental $^{87}$Sr/$^{86}$Sr values. For example, our observed Monte Albán plant $^{87}$Sr/$^{86}$Sr values ranged from 0.707607 to 0.708005, with a mean of $^{87}$Sr/$^{86}$Sr = 0.707809 ± 0.000139 (1$\sigma$, $n = 8$). Though wider, the BART model produces a local range calibrated to actual geologic and environmental input strontium sources that more accurately capture $^{87}$Sr/$^{86}$Sr cycling within the environment rather than arbitrarily imposed statistical thresholds independent of the broader environmental context. Furthermore, the predicted local range from our isoscape model provides additional benefits over traditional statistical biogeochemical baselines generated from observed environmental $^{87}$Sr/$^{86}$Sr values. Whereas we collected plant samples opportunistically (and therefore idiosyncratically) over a limited portion of the site, the isoscape provides a geographically standardized estimated local $^{87}$Sr/$^{86}$Sr range for Monte Albán over a 1 km radius, allowing us to more accurately characterize local strontium values.

Even so, the isoscape's predicted $^{87}$Sr/$^{86}$Sr range represents only a conservative approximation of "local" Monte Albán $^{87}$Sr/$^{86}$Sr values. Our predicted local range is based on a single 1 km cell from the isoscape model. However, an archaeological survey of Monte Albán shows that the site was approximately 3 sq km in size [14]. While Nicholas and Feinman propose 1–2 km as a reasonable catchment area for agricultural subsistence for most prehispanic settlements within the Valley of Oaxaca, they conclude that Monte Albán likely drew from beyond a 2 km radius to meet its subsistence needs [24]. While we cannot know from exactly where Monte Albán drew its food supply—and thus its residents' $^{87}$Sr/$^{86}$Sr intake—the wider but imperfect 1 km predicted isoscape range certainly represents a more realistic characterization of expected local $^{87}$Sr/$^{86}$Sr values in ancient residents of Monte Albán than the narrower local range based solely on modern plant samples.

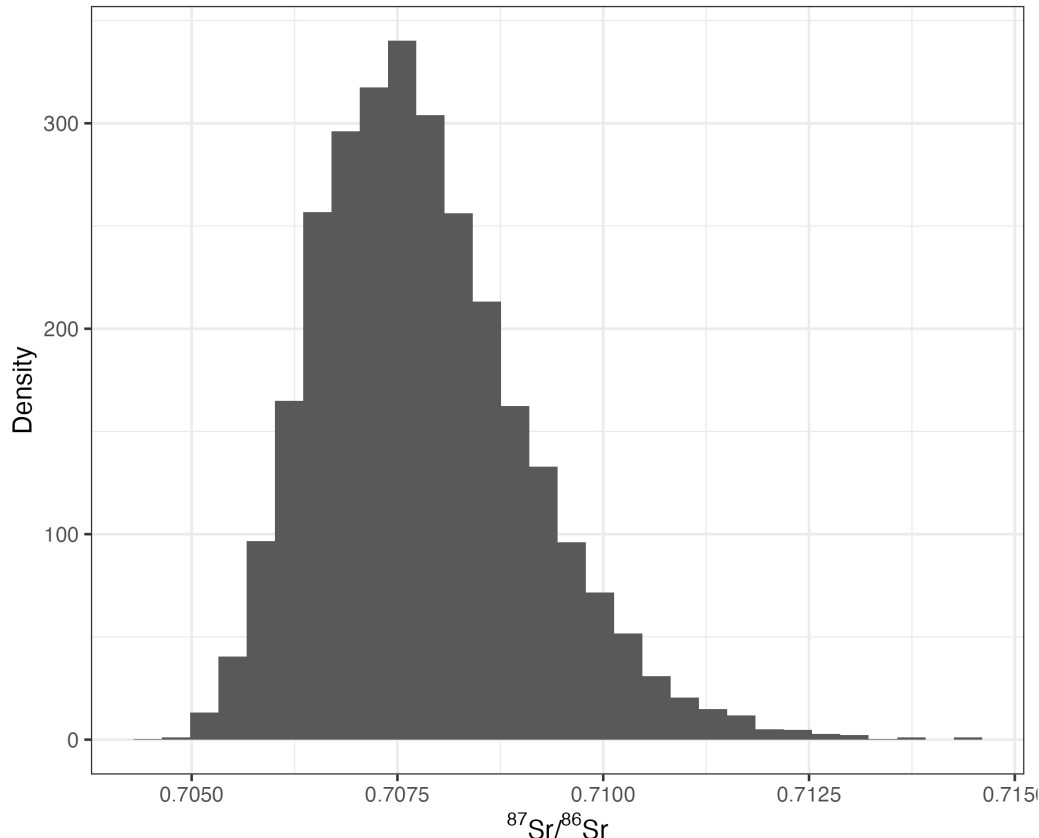

**Fig 6. Posterior predictive distribution of expected "local" $^{87}$Sr/$^{86}$Sr at Monte Albán from the final BART model.**

When we apply the isoscape predicted local range to previously published Monte Albán human $^{87}$Sr/$^{86}$Sr values and their associated standard errors, we come away with a more nuanced understanding of paleomobility at the site (Fig 7). While individuals previously categorized as locals ($n = 4$) and non-locals ($n = 1$) comfortably remain in these categories [10], their sampled tissues' placement either well within or on the margins of the Monte Albán predicted local range gives us a sense of the relative certainty or uncertainty of these designations. For example, if the standard error associated with a "local" enamel or bone $^{87}$Sr/$^{86}$Sr value extended beyond the predicted Monte Albán local range, this would indicate greater uncertainty in the residential designation of that individual. This is not the case among Monte Albán individuals, where all previously published $^{87}$Sr/$^{86}$Sr values have extremely low associated standard errors (<0.00002). These error ranges are smaller than the pixel size of the points themselves and therefore are not visualized in the figure. This indicates that each of these tissues can confidently be assigned a "local" or "non-local" residential status.

Moreover, nearly all "local" tissues (i.e., sampled tissues whose observed $^{87}$Sr/$^{86}$Sr values fall within the predicted local Monte Albán range) exhibit $^{87}$Sr/$^{86}$Sr values that cluster closely to the median isoscape predicted "local" Monte Albán value of $^{87}$Sr/$^{86}$Sr = 0.7077. Burial 26 A's femoral $^{87}$Sr/$^{86}$Sr measurement represents the sole exception among "local" tissues, falling near the lower bounds of the predicted local Monte Albán range. This is still consistent with an individual who lived locally in the last ~10 years of life. However, a bone $^{87}$Sr/$^{86}$Sr values nearer to the margins of the predicted local range could indicate that this individual relocated to Monte Albán recently in later life and that their bone $^{87}$Sr/$^{86}$Sr values recently acclimated to local $^{87}$Sr/$^{86}$Sr values at the time of death. We thus get a better sense of the certainty of our designation of

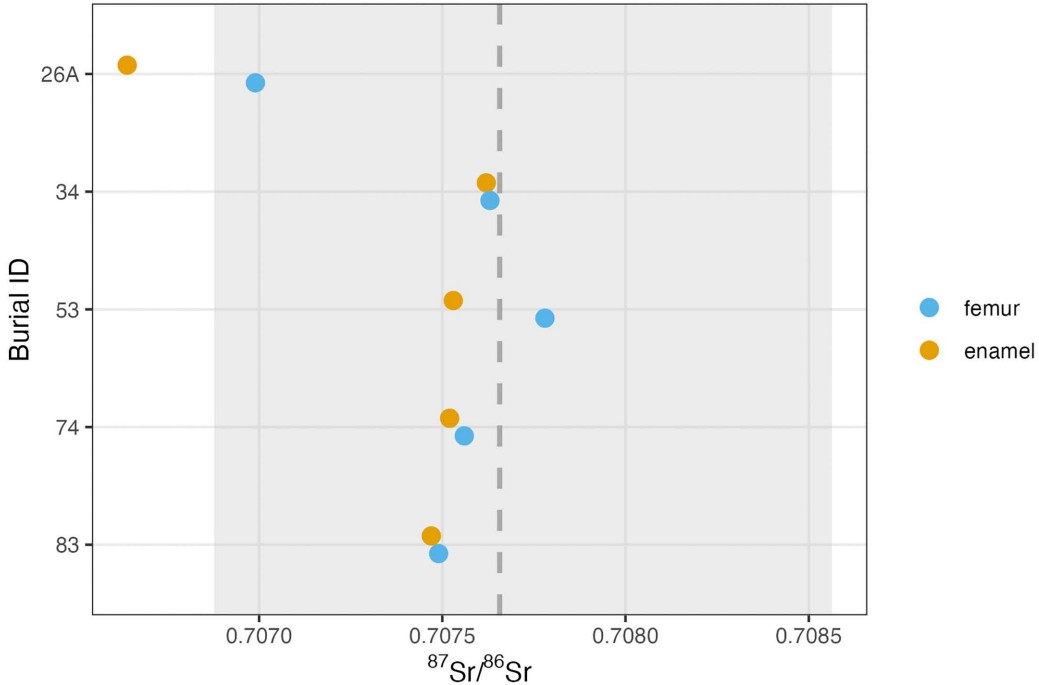

**Fig 7. Previously published paired human enamel and bone samples from Monte Albán compared to the BART predictions at Monte Albán.** The grey dashed line indicates the median BART $^{87}$Sr/$^{86}$Sr prediction for Monte Albán. The grey shaded area indicates the interquartile range of BART predictions for local Monte Albán $^{87}$Sr/$^{86}$Sr values. Reported standard error for each observed $^{87}$Sr/$^{86}$Sr value was sufficiently low (<0.00002) that error bars were not included in the graph [10].

"local" and "non-local" status among individuals interred at Monte Albán, recognizing that biogeochemical locality (i.e., falling within the Monte Albán predicted local range) does not necessarily equate cultural locality.

## Conclusion

We created a BART predictive $^{87}$Sr/$^{86}$Sr isoscape model of Mesoamerica and the Valley of Oaxaca, integrating novel and compiled empirical Mesoamerican $^{87}$Sr/$^{86}$Sr data in addition to geological maps and geoenvironmental Sr covariate spatial data. The BART model's improved performance over a global $^{87}$Sr/$^{86}$Sr isoscape model based on geologic bedrock alone indicates both the importance of regional calibration in developing predictive isoscapes, as well as the importance of accounting for the varying influence of geoenvironmental factors such as climate and soil processes on bioavailable $^{87}$Sr/$^{86}$Sr. We anticipate that the model will be further improved with the incorporation of additional empirical $^{87}$Sr/$^{86}$Sr values from across greater Mesoamerica, particularly in regions with high native uncertainty estimates.

Our isoscape indicates that clear $^{87}$Sr/$^{86}$Sr variation exists within greater Mesoamerica and the Valley of Oaxaca. Although there is some overlap in $^{87}$Sr/$^{86}$Sr values across the three branches of the Valley of Oaxaca, there is sufficient variation between them to detect intra-valley migration, as well as migration between the Valley and greater Mesoamerica. By providing a high performing predictive $^{87}$Sr/$^{86}$Sr isoscape for the Valley, we hope that biogeochemical investigations of migration in the region will expand beyond their initial focus on the Zapotec state capital of Monte Albán. The inclusion of all three branches of the Valley in the isoscape will allow researchers to consider questions about the scale of migration in the Valley and what role both intra- and interregional migration played in the formation and maintenance of the Zapotec state, as well as in the subsequent cultural development of the region.

Finally, the isoscape provides archaeologists with a more refined tool for the examination and discussion of past migratory patterns within ancient Mesoamerica. The BART model's standardized and geoenvironmentally informed predictions of expected "local" $^{87}Sr/^{86}Sr$ ranges for archaeological sites of interest provides clear advantages over the use of traditional statistical threshold-based approaches that do not account for the broader environmental context's effects on local $^{87}Sr/^{86}Sr$ variability. Moreover, the model's explicit quantification of uncertainty in predicted $^{87}Sr/^{86}Sr$ values provides archaeologists with crucial information about confidence in expected "local" $^{87}Sr/^{86}Sr$ ranges that directly impacts the reliability of migrant identification.

## Supporting information

**S1 Checklist. Inclusivity in global research checklist.**
(PDF)

**S1 Fig. PCA results identifying the major environmental factors driving strontium isotope patterns in Mesoamerica.**
(TIF)

**S2 Fig. Upper and lower 95% confidence intervals for $^{87}Sr/^{86}Sr$ prediction ranges in Mesoamerican isoscape.** Map created with publicly available country and Mexican state boundaries from Natural Earth [86].
(TIF)

**S1 Appendix. Identification of contamination from agricultural fertilizers in modern plant samples from San Martín Tilcajete.**
(PDF)

**S2 Appendix. Spanish translation of manuscript.**
(PDF)

## Acknowledgments

We are grateful to Pedro Fabian Ojeda for assistance in collecting plant samples in the Ocotlán-Zimatlán arm of the Valley of Oaxaca. At Arizona State University's Archaeological Chemistry Laboratory, we are thankful to research apprentices Eric Flores, Kari Guilbault, Sparshee Naik, and Tajinder Virdee for assistance with sample processing. At the Metals, Environmental, and Terrestrial Analytical Laboratory, we are grateful for the assistance of Dr. Stephen Romaniello, Trevor Martin, and Natasha Zolotova.

## Author contributions

**Conceptualization:** Sofía I. Pacheco-Forés, Nicolas Gauthier.

**Data curation:** Nicolas Gauthier.

**Formal analysis:** Sofía I. Pacheco-Forés, Nicolas Gauthier.

**Funding acquisition:** Sofía I. Pacheco-Forés, Nicolas Gauthier, Lacey B. Carpenter.

**Methodology:** Gwyneth Gordon, Kelly J. Knudson.

**Project administration:** Sofía I. Pacheco-Forés.

**Resources:** Lacey B. Carpenter, Gwyneth Gordon, Kelly J. Knudson.

**Visualization:** Nicolas Gauthier.

**Writing – original draft:** Sofía I. Pacheco-Forés, Nicolas Gauthier.

**Writing – review & editing:** Sofía I. Pacheco-Forés, Lacey B. Carpenter, Gwyneth Gordon, Kelly J. Knudson.

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
