## [Decision Letter · Decision Letter 0]

13 Jun 2025

Dear Dr. Pacheco-Forés,

Thank you for submitting your manuscript to PLOS ONE. After careful consideration, we feel that it has merit but does not fully meet PLOS ONE’s publication criteria as it currently stands. Therefore, we invite you to submit a revised version of the manuscript that addresses the points raised during the review process.

1) I agree with your point that standard Random Forest (RF) models do not inherently provide uncertainty estimates. However, specific extensions of RF, such as Quantile Regression Forests, have been developed to address this limitation. In the paper, you reference two seminal works by Bataille and co-authors that present isoscape maps without including spatial uncertainty. However, more recent studies have explicitly addressed this issue. Therefore, I recommend a more detailed review of the isoscape literature—particularly with regard to machine learning approaches—and the inclusion of relevant and recent publications that account for uncertainty in spatial predictions.

We look forward to receiving your revised manuscript.

Kind regards,

Federico Lugli, Ph.D.

Academic Editor

PLOS ONE

Journal Requirements:

4. We note that Figures 1,2,3 and 4 in your submission contain [map/satellite] images which may be copyrighted. All PLOS content is published under the Creative Commons Attribution License (CC BY 4.0), which means that the manuscript, images, and Supporting Information files will be freely available online, and any third party is permitted to access, download, copy, distribute, and use these materials in any way, even commercially, with proper attribution. For these reasons, we cannot publish previously copyrighted maps or satellite images created using proprietary data, such as Google software (Google Maps, Street View, and Earth). For more information, see our copyright guidelines: http://journals.plos.org/plosone/s/licenses-and-copyright.

1. You may seek permission from the original copyright holder of Figures 1,2,3 and 4 to publish the content specifically under the CC BY 4.0 license. 

I request permission for the open-access journal PLOS ONE to publish XXX under the Creative Commons Attribution License (CCAL) CC BY 4.0 (http://creativecommons.org/licenses/by/4.0/). Please be aware that this license allows unrestricted use and distribution, even commercially, by third parties. Please reply and provide explicit written permission to publish XXX under a CC BY license and complete the attached form.

Reviewers' comments:

Reviewer's Responses to Questions

**Comments to the Author**

1. Is the manuscript technically sound, and do the data support the conclusions?

Reviewer #1: Yes

Reviewer #2: Yes

2. Has the statistical analysis been performed appropriately and rigorously?

Reviewer #1: Yes

Reviewer #2: Yes

3. Have the authors made all data underlying the findings in their manuscript fully available?

Reviewer #1: Yes

Reviewer #2: Yes

4. Is the manuscript presented in an intelligible fashion and written in standard English?

Reviewer #1: Yes

Reviewer #2: Yes

Reviewer #1: In this manuscript, the authors used a machine learning approach, BART, to generate a bioavailable 87Sr/86Sr isoscape for Oaxaca Valley and Mesoamerica. This study contributes to the 87Sr/86sr mapping of a region poorly documented compared to other of Mesoamerica or Europe or North America.

I enjoyed the reading of the paper because it is well written, and the model really interesting. I appreciated the case study at the end, which effectively highlights the potential of the isoscape for provenance study in Oaxaca Valley.

I do not have major comments, I only have suggestions that may help further improve the manuscript:

1) Table 2: add errors to Sr isotopic ratios.

2) The geological context of the Oaxaca Valley is a little poor. Expanding it and also discussing the isotopic data related to the geology could help to better constrain the isotopic variation found in the area. A geological map of the Oaxaca Valley could be useful to understand the geological complexity of the region.

3) Why don't you tried to make the model using only the Mesoamerican and Oaxaca valley data, being the North and South America data extremely far?

4) Some locations you have sampled show a huge isotopic range as, for example, the Monte Albán, with variation of 0.00055. How do you explain this?

5) As for the example you provide of Monte Albán, the isoscape prediction of a range from 0.7069 to 0.7086 is also huge. Is this range calculated for a singe 1 km cell or from a larger area? Is it related to a particular geological complexity of the site or from other factors?

6) If my interpretation is correct, the 'measurement' of archaeological literature samples you use to test the model from Albàn site and reported in Fig. 6 as boxplot were simulated. I doubt that during the single analysis, the 87Sr/86Sr isotopic ratio varies as much as you report. I suggest you to use only the published values and associated errors in the absence of raw data.

7) In Fig. 2 and 3 write 87Sr/86Sr and not Sr on the legend.

Reviewer #2: Congratulations. I enjoyed reading and reviewing this manuscript.

There are some minor edits as comments on the annotated pdf.

I understand the complexities of running isoscape models to produce the maps presented in the manuscript, and this is a large hurdle for non-experts wanting to use constructed isoscapes. You present the new data, but not the compiled data used. Also, I do not see the scripts used for the modeling included. These are invaluable for future researchers hoping to similarly construct isoscapes and use these following your approach. Could these be include as supplementary information?

**Do you want your identity to be public for this peer review?** For information about this choice, including consent withdrawal, please see our Privacy Policy

Reviewer #1: No

Reviewer #2: **Yes: ** Petrus le Roux

---

## [Author Response · Author response to Decision Letter 1]

15 Oct 2025

Academic Editor Comments

1. I agree with your point that standard Random Forest (RF) models do not inherently provide uncertainty estimates. However, specific extensions of RF, such as Quantile Regression Forests, have been developed to address this limitation. In the paper, you reference two seminal works by Bataille and co-authors that present isoscape maps without including spatial uncertainty. However, more recent studies have explicitly addressed this issue. Therefore, I recommend a more detailed review of the isoscape literature—particularly with regard to machine learning approaches—and the inclusion of relevant and recent publications that account for uncertainty in spatial predictions.

We have expanded our isoscape literature review section to include recent work on the use of quantile regression forests to quantify spatial uncertainty in isoscapes. We have also referenced this approach in our discussion of the advantages of our BART approach to isoscape modeling, primarily in its handling of uncertainty in data-sparse regions.

2. In Figure 6, you present simulated draws based on the mean and standard deviation of published data, which are then compared to the expected local baseline from the isoscape. I believe it would be important to clearly explain the rationale and methodology behind generating these simulated data. This is especially relevant since the isotopic variability represented by a single sample appears to be significantly larger than any analytical uncertainty. Additionally, it is unclear why you did not simply use the average value along with its associated standard error, which might offer a more straightforward and transparent comparison.

We have edited this figure to use the published 87Sr/86Sr value and its associated standard error rather than the simulated data.

Reviewer 1 Comments

1. Table 2: add errors to Sr isotopic ratios.

We have added 2 SE to each 87Sr/86Sr observation in Table 2.

2. The geological context of the Oaxaca Valley is a little poor. Expanding it and also discussing the isotopic data related to the geology could help to better constrain the isotopic variation found in the area. A geological map of the Oaxaca Valley could be useful to understand the geological complexity of the region.

We have expanded our discussion of the geological context of the Valley of Oaxaca and integrated discussions of the geology into our discussion of isoscape 87Sr/86Sr predictions across the Valley. Additionally, we have included a geological map of the Valley of Oaxaca in manuscript (Fig 2).

3. Why don't you try to make the model using only the Mesoamerican and Oaxaca valley data, being the North and South America data extremely far?

We tested models trained exclusively on Mesoamerican data and found essentially identical performance on held-out Oaxacan samples compared to models including all North and South American data. However, we retained the continental-scale training data because geographically distant sites are not necessarily distant in environmental feature space—a Precambrian terrain in Brazil may be more similar to parts of the Oaxacan highlands than geographically proximate volcanic areas. The expanded dataset provides greater environmental parameter coverage without degrading local performance, potentially improving model robustness for out-of-sample predictions across the full range of geological and climatic conditions relevant to archaeological applications in our study region.

4. Some locations you have sampled show a huge isotopic range as, for example, the Monte Albán, with variation of 0.00055. How do you explain this?

Local 87Sr/86Sr can vary a great deal even within a relatively small geographical area. Countless studies, as well as our own isoscape model (S1 Fig), show that the age of the underlying geologic bedrock is the primary driver of variation in bioavailable 87Sr/86Sr. Given that the Valley of Oaxaca is located in the most geologically complex region of Mexico and contains some of the oldest Precambrian formations in Mexico that are overlain with isolated Paleozoic, Mesozoic, and Cenozoic formations, it is expected that there would be greater variability in local 87Sr/86Sr values, even within sites.

Furthermore, local bioavailable 87Sr/86Sr values are also influenced by environmental factors such as differential weathering and the addition of atmospheric sediments such as windblown dust, volcanic tephra, and aerosolized sea salt in sea spray (see reviews in Bentley 2006; Blum et al, 2000). Attempting to capture this variation through plants adds an additional layer of variation, as not all 87Sr/86Sr is bioavailable to plants, and plant rooting depths determine what proportion of bedrock vs environmental 87Sr/86Sr will make up a plant’s own 87Sr/86Sr value (Stewart et al., 1998; Poszwa et al., 2002; Poszwa et al., 2004). For this reason, we sampled plants of various rooting depths to more accurately represent local bioavailable 87Sr/86Sr.

Despite these sources of expected variation, Monte Albán’s local range of 87Sr/86Sr variation is in line with the majority of the sites sampled in the study. The 16 sites across the Valley of Oaxaca demonstrate an average local 87Sr/86Sr range of 0.000482. This average excludes San Martín Tilcajete, whose samples showed a 87Sr/86Sr range an order of magnitude larger (0.004739) due to contamination from agricultural fertilizer use (discussed in S1 Appendix).

Local 87Sr/86Sr ranges of ~0.00050 are also fairly typical elsewhere in Mesoamerica. In a study characterizing baseline 87Sr/86Sr variability in central Mexico through analysis of environmental samples (plants, soils, water, fauna), Pacheco-Forés et al (2020) found a similar average local 87Sr/86Sr range of 0.00046 in 13 sites across central Mexico. Similarly, in a study of 87Sr/86Sr variation at the site of Copán in Honduras, Price et al (2010) found a local 87Sr/86Sr range of ~0.00040 (plants, water, fauna).

While the Valley of Oaxaca site 87Sr/86Sr ranges thus may be slightly higher than those observed elsewhere in Mesoamerica, the ranges are not so large as to call into question the validity of the observed 87Sr/86Sr values nor to imply they could be contaminated with agricultural fertilizers as the San Martín Tilcajete samples were.

Bentley, R. Alexander. “Strontium Isotopes from the Earth to the Archaeological Skeleton: A Review.” Journal of Archaeological Method and Theory 13, no. 3 (July 29, 2006): 135–87. https://doi.org/10.1007/s10816-006-9009-x.

Blum, Joel D., E. Hank Taliaferro, Marie T. Weisse, and Richard T. Holmes. “Changes in Sr/Ca, Ba/Ca and 87Sr/86Sr Ratios between Trophic Levels in Two Forest Ecosystems in the Northeastern U.S.A.” Biogeochemistry 49 (2000): 87–101.

Pacheco-Forés, Sofía I., Gwyneth W. Gordon, and Kelly J. Knudson. “Expanding Radiogenic Strontium Isotope Baseline Data for Central Mexican Paleomobility Studies.” PLOS ONE 15, no. 2 (February 24, 2020): e0229687. https://doi.org/10.1371/journal.pone.0229687.

Poszwa, Anne, Etienne Dambrine, Bruno Ferry, Benoît Pollier, and Michel Loubet. “Do Deep Tree Roots Provide Nutrients to the Tropical Rainforest?” Biogeochemistry 60, no. 1 (August 1, 2002): 97–118. https://doi.org/10.1023/A:1016548113624.

Poszwa, Anne, Bruno Ferry, Etienne Dambrine, Benoît Pollier, Tonie Wickman, Michel Loubet, and Kevin Bishop. “Variations of Bioavailable Sr Concentration and 87Sr/86Sr Ratio in Boreal Forest Ecosystems.” Biogeochemistry 67, no. 1 (January 1, 2004): 1–20. https://doi.org/10.1023/B:BIOG.0000015162.12857.3e.

Price, T. Douglas, James H. Burton, Robert J. Sharer, Jane E. Buikstra, Lori E. Wright, Loa P. Traxler, and Katherine A. Miller. “Kings and Commoners at Copan: Isotopic Evidence for Origins and Movement in the Classic Maya Period.” Journal of Anthropological Archaeology 29, no. 1 (March 2010): 15–32. https://doi.org/10.1016/j.jaa.2009.10.001.

Stewart, Brian W, Rosemary C Capo, and Oliver A Chadwick. “Quantitative Strontium Isotope Models for Weathering, Pedogenesis and Biogeochemical Cycling.” Geoderma 82, no. 1 (February 1, 1998): 173–95. https://doi.org/10.1016/S0016-7061(97)00101-8.

5. As for the example you provide of Monte Albán, the isoscape prediction of a range from 0.7069 to 0.7086 is also huge. Is this range calculated for a single 1 km cell or from a larger area? Is it related to a particular geological complexity of the site or from other factors?

We have clarified that the isoscape’s predicted local range for Monte Albán is calculated for a single 1 km cell. The 87Sr/86Sr range of this 1 km cell is indeed fairly large (0.00170), especially when compared to the 87Sr/86Sr range observed in modern plant samples from Monte Albán (0.00055). However, as we discussed in the manuscript, the isoscape’s wider predicted local 87Sr/86Sr range for Monte Albán more accurately reflects actual geologic and environmental inputs to local 87Sr/86Sr values as opposed to the idiosyncratic and opportunistic sampling of modern plants across a limited subsection of the contemporary archaeological site that produced an artificially narrow local 87Sr/86Sr range. The isoscape takes into account the age and composition of the underlying geologic bedrock as well as environmental Sr inputs, including the overall aridity, dry vs wet atmospheric aerosol deposition, elevation, and other factors. These factors, along with others that drive the isoscape model’s baseline predictions, are summarized in the supplemental document S1 Fig.

Additionally, the isoscape’s predicted 87Sr/86Sr range represents a conservative approximation of “local” Monte Albán 87Sr/86Sr values. Although we based our prediction on a single 1km cell from the model, archaeological surveys of the site show that it was approximately 3 sq km in size (Blanton 1978: Fig 1.3). While Nicholas and Feinman (2022) propose 1-2km as a reasonable catchment area for agricultural subsistence for most prehispanic settlements within the Valley of Oaxaca, they conclude that Monte Albán likely drew from beyond a 2km radius to fully meet its subsistence needs (p10). While we cannot know from where exactly Monte Albán was drawing its food supply (and thus its residents 87Sr/86Sr intake), the wider 1 km predicted isoscape range certainly represents a more realistic characterization of expected local 87Sr/86Sr values in ancient residents of Monte Albán than the narrower local range based solely on modern plant samples.

Blanton, Richard E. Monte Albán: Settlement Patterns at the Ancient Zapotec Capital. New York, NY: Academic Press, 1978.

Nicholas, Linda M., and Gary M. Feinman. “The Foundation of Monte Albán, Intensification, and Growth: Coactive Processes and Joint Production.” Frontiers in Political Science 4 (March 8, 2022): 805047. https://doi.org/10.3389/fpos.2022.805047.

6. If my interpretation is correct, the 'measurement' of archaeological literature samples you use to test the model from Albàn site and reported in Fig. 6 as boxplot were simulated. I doubt that during the single analysis, the 87Sr/86Sr isotopic ratio varies as much as you report. I suggest you to use only the published values and associated errors in the absence of raw data.

We have edited this figure to use the published 87Sr/86Sr value and its associated standard error rather than the simulated data.

7. In Fig. 2 and 3 write 87Sr/86Sr and not Sr on the legend.

Fixed.

Reviewer 2 Comments

1. I understand the complexities of running isoscape models to produce the maps presented in the manuscript, and this is a large hurdle for non-experts wanting to use constructed isoscapes. You present the new data, but not the compiled data used. Also, I do not see the scripts used for the modeling included. These are invaluable for future researchers hoping to similarly construct isoscapes and use these following your approach. Could these be included as supplementary information?

We have provided both the compiled data used in the model as well as the scripts for the modeling here: https://github.com/flmnh-ai/oaxaca-isoscape. We included this in the data availability metadata when we submitted our manuscript and will confirm with the editor that the link will appear in the final publication.

2. Line 27: which regions?

We have specified that these two regions are the Teotihuacan Valley and the Maya region.

3. Line 212: typo “some of”

Corrected.

4. Line 384 “Isoscape model performance”: for non-expert readers, it would be good to quickly define what “good” looks like in terms of RMSE and R2. Many people interested in isoscapes are not experts in statistics but understanding these performance evaluations is critical.

We have added additional context to the methods and results sections to clarify interpretation of the RMSE and R2 metrics.

5. Fig 2 & 3: “87Sr/86Sr” should be in the legend of the figure, not “Sr” which is confusing and might be interpreted to mean elemental Sr abundance. At present this doesn’t make sense.

Fixed.

6. Fig 4: The legend should indicate–or the figure caption–the the CI intervals in the legend are in terms of 87Sr/86Sr

Fixed.

7. Fig 5 & 6: “87Sr/86Sr” in the x-axis label

Fixed.

---

## [Editor Report · Decision Letter 1]

2 Nov 2025

Dear Dr. Pacheco-Forés,

I reviewed the paper myself, and I thank the authors for having revised the manuscript based on the referees' comments. I still have two minor (but, I believe, important) points that need to be addressed before publication.

First of all, I kindly ask the authors to tone down the sentence at line 216 to something like: "this approach may underestimate uncertainty in regions...".

We look forward to receiving your revised manuscript.

Kind regards,

Federico Lugli, Ph.D.

Academic Editor

PLOS ONE
---

## [Author Response · Author response to Decision Letter 2]

11 Nov 2025

Thank you for these additional revisions. We have edited the sentence at line 216 to read "this approach may underestimate uncertainty..." as requested.

In addition, we have updated Figure 7 to accurately reflect the SE data published in Price et al 2000 Table 1. The corrected SE ranges were so small that it was not possible to visualize them on the graph as they were smaller than the pixel size of the point itself in Figure 7. We have revised the figure caption, as well as the paragraphs in our Discussion interpreting the figure to reflect this change. We are incredibly thankful to the reviewer for catching this error on our part!

---

## [Editor Report · Decision Letter 2]

25 Nov 2025

Radiogenic strontium isotope variability in the Valley of Oaxaca: A predictive isoscape for Mesoamerican paleomobility studies

PONE-D-25-23485R2

Dear Dr. Pacheco-Forés,

We’re pleased to inform you that your manuscript has been judged scientifically suitable for publication and will be formally accepted for publication once it meets all outstanding technical requirements.

Kind regards,

Federico Lugli, Ph.D.

Academic Editor

PLOS ONE
---

## [Editor Report · Acceptance letter]

2 Dec 2025

PONE-D-25-23485R2

PLOS ONE

Dear Dr. Pacheco-Forés,

I'm pleased to inform you that your manuscript has been deemed suitable for publication in PLOS ONE. Congratulations! Your manuscript is now being handed over to our production team.

Kind regards,

on behalf of

Dr. Federico Lugli

Academic Editor

PLOS ONE